

# Evaluation of a multi-model, multi-constituent assimilation framework for tropospheric chemical reanalysis

Kazuyuki Miyazaki[1,2], Kevin W. Bowman[1], Keiya Yumimoto[3], Thomas Walker[4], Kengo Sudo[5,1]

[1]Jet Propulsion Laboratory, California Institute of Technology, Pasadena, CA, USA
5 [2]Earth Surface System Research Center, Japan Agency for Marine-Earth Science and Technology (JAMSTEC), Yokohama, 236-0001, Japan
[3]Research Institute for Applied Mechanics, Kyushu University, Kasuga Park 6-1, Fukuoka, 816-8580, Japan
[4]Department of Civil and Environmental Engineering, Carleton University, Ottawa, Ontario, Canada
[5]Graduate School of Environmental Studies, Nagoya University, Nagoya, Japan

*Correspondence to*: Kazuyuki Miyazaki (kazuyuki.miyazaki@jpl.nasa.gov)

**Abstract.** We introduce a Multi-mOdel Multi-cOnstituent Chemical data assimilation (MOMO-Chem) framework that directly accounts for model error in transport and chemistry by integrating a portfolio of forward chemical transport models (GEOS-Chem, AGCM-CHASER, MIROC-Chem, MIROC-Chem-H) into a state-of-the-art ensemble Kalman filter data 15 assimilation system that simultaneously optimizes both concentrations and emissions of multiple species through ingestion of a suite of measurements (ozone, $NO_2$, CO, $HNO_3$) from multiple satellite sensors. In spite of substantial model differences, the observational density and accuracy was sufficient for the assimilation to reduce the multi-model spread by 20–85% for ozone, and annual mean bias by 39–97% for ozone in the middle troposphere, while simultaneously reducing the tropospheric $NO_2$ column biases by more than 40%, and the negative biases of surface CO in the Northern Hemisphere by 20 41–94%. For tropospheric mean OH, the multi-model mean meridional hemispheric gradient was reduced from $1.32\pm0.03$ to $1.19\pm0.03$, while the multi-model spread was reduced by 24–58% over polluted areas. These improvements extended to emissions where uncertainty ranges in the a posteriori emissions due to model errors were quantified in 4–31% for NOx and 13–35% for CO regional emissions. Harnessing assimilation increments in both NOx and ozone, we show that the sensitivity of ozone and $NO_2$ surface concentrations to NOx emissions varied by a factor of 2 for end-member models revealing 25 fundamental differences in the representation of fast chemical and dynamical processes. Consequently, diagnostic information readily available from MOMO-Chem has the potential to improve chemical predictions through relationships such as emergent constraints.



# 1 Introduction

Data assimilation is a technique for combining different observational data sets with a model, taking into consideration of the characteristics of individual measurements and model dynamics (e.g., Kalnay, 2003; Lahoz and Schneider, 2014). Atmospheric composition and chemical data assimilation using advanced data assimilation techniques such as four-dimensional variational data assimilation (4D-VAR) and ensemble Kalman filter (EnKF) allow the propagation of observational information in time and space from a limited number of observed species to a wide range of chemical components (e.g., Lahoz et al, 2007; Sandu et al., 2011; Bocquet et al., 2015). Data assimilation provides global fields that are statistically consistent with individual observations. Various studies have demonstrated the capabilities of chemical data assimilation systems in the analysis of chemical species in the troposphere and stratosphere (e.g. Parrington et al., 2009; Kiesewetter et al., 2010; Flemming et al., 2011; Coman et al., 2012; Emili et al., 2014; Miyazaki et al., 2012a, 2012b, 2015, 2019; van der A et al., 2015), emissions optimization (e.g., Miyazaki et al., 2012a, 2013, 2014; Stavrakou et al. 2013; Streets et al., 2013; Inness et al., 2015; Jiang et al., 2018), and chemical reanalyses to provide long-term data assimilation products (Inness et al., 2013; Gaubert et al., 2016; Miyazaki et al., 2015; Flemming et al., 2017). Chemical data assimilation frameworks have also been used to evaluate observing system impacts through observation system simulation experiments (OSSEs) (Yumimoto, 2013; Lahoz and Schneider, 2014; Bocquet et al., 2015; Abida et al., 2017; Liu et al., 2017) and evaluate chemistry-climate model simulations (Miyazaki and Bowman, 2017; Kuai et al., 2019).

Developments of advanced data assimilation techniques and satellite retrievals have contributed to improving data assimilation analysis and prediction of atmospheric composition (e.g., Singh et al., 2012; Skachko et al., 2016; Boersma et al., 2018). However, a limiting factor in the accuracy of these systems is the performance of forecast models, that have limited fidelity in the representation of atmospheric dynamics and chemistry. For example, intercomparison studies of the Atmospheric Chemistry and Climate Model Intercomparison Project (ACCMIP) (Bowman et al., 2013; Young et al., 2013; Stevenson et al., 2013) and the Chemistry-Climate Model Initiative (CCMI) (Morgenstern et al., 2018; Kuai et al., 2019) revealed a large diversity in simulations of tropospheric composition owing to differences in model processes and input data. The choice of forecast model, thus, largely influences the a priori uncertainty in chemical data assimilation and the a posteriori data assimilation analysis.

As opposed to 4D-Variational techniques that requires a model adjoint, EnKF systems are independent from forecast model code and therefore can readily integrate multiple models into a multi-model data assimilation framework (Houtekamer and Zhang, 2016). EnKF techniques have been successfully applied to multiple different chemical transport models (CTMs) in our previous studies (e.g., Miyazaki et al., 2012, 2015, 2017, 2019), which have been used to assimilate multi-constituent composition measurements from multiple sensors where both the chemical states and emissions of various species were simultaneously optimized. However, the sensitivity of concentrations to emissions, such as ozone response to NOx emissions, is strongly model dependent and therefore has a first-order impact on the performance in multi-constituent data assimilation framework. Consequently, quantification of this impact is important not only for analysis but also for Observing

System Simulation Experiments (OSSEs) used to assess and design new observing systems. Nevertheless, the importance of forecast model performance on chemical data assimilation has not been demonstrated using a common data assimilation framework for tropospheric chemistry analysis. A multi-model framework can also be used to provide multi-model integrated analysis fields, which are less dependent on individual model performance.

This study demonstrates, for the first time, the importance of forecast model performance on data assimilation analysis of tropospheric composition and emissions, by utilizing four different CTM frameworks and applying a common EnKF approach. For this, an EnKF data assimilation system based on the GEOS-Chem model is newly developed in this study. Using the same data assimilation settings and assimilating multi-constituent observations from multiple satellite sensors, we examine how model bias affects tropospheric chemistry data assimilation performance, including emission estimation, and

provide integrated data assimilation analysis fields from an ensemble of analyses that ingested multiple models and multi-constituent measurements.

## 2 Methodology

### 2.1 Data assimilation module

The data assimilation technique is based upon on a Local Ensemble Transform Kalman Filter (LETKF) approach

developed by Hunt et al. (2007). The LETKF uses an ensemble forecast to estimate the background error covariance matrix and generates an analysis ensemble mean and covariance that satisfy the Kalman filter equations for linear models. In the forecast step, a background ensemble, $x_i^b$ (i=1,...,k), is obtained from the evolution of an ensemble model forecast. Here, $\mathbf{x}$ represents the model variable, b indicates the background state, and k is the ensemble size (32 in this study). The background ensemble mean $\overline{x^b}$ and its perturbation $X^b$ are then estimated as follows:

$$\overline{x^b} = \frac{1}{k}\sum_{i=1}^{k} x_i^b \quad (1)$$

$$X_i^b = x_i^b - \overline{x^b} \quad (2)$$

The background error covariance is then estimated at each time step at each grid point as follows:

$$P^b = X^b(X^b)^T \quad (3)$$

The background ensemble is converted into the observation space, $y_i^b = H(x_i^b)$, using the observation operator $H$, which is

the composite of a spatial interpolation operator and a satellite retrieval operator (c.f., Section 2.3). An ensemble of background perturbation is defined as $Y_i^b = y_i^b - \overline{y^b}$.

Using the covariance matrices of observation and background error, the data assimilation determines the relative weights of the observation and background, and subsequently transforms a background ensemble into an analysis ensemble, $x_i^a$ (i=1,...,k). The analysis ensemble mean $\overline{x^a}$ is obtained by updating the background ensemble mean as follows:

$$\overline{x^a} = \overline{x^b} + X^b\widetilde{P}^a(Y^b)^T R^{-1}(y^o - \overline{y^b}) \quad (4)$$



$$\widetilde{\boldsymbol{P}}^a = \left[\frac{(k-1)}{1+\Delta}\boldsymbol{I} + \left(\boldsymbol{Y}^b\right)^T\boldsymbol{R}^{-1}\boldsymbol{Y}^b\right]^{-1} (5)$$

where $\widetilde{\boldsymbol{P}}^a$ is the $k \times k$ local analysis error covariance in the ensemble space, $\boldsymbol{y}^o$ is the observation vector, and $\boldsymbol{R}$ is the observation error covariance. A covariance inflation factor ($\Delta$) is applied to inflate the forecast error covariance.

The observation-minus-forecast (OmF), that is known as the observational increment, is defined as:

$$\boldsymbol{y}^o - \overline{\boldsymbol{y}^b} (6)$$

The analysis increment is defined as the correction made by data assimilation as follows:

$$\overline{\boldsymbol{x}^a} - \overline{\boldsymbol{x}^b} (7)$$

The analysis ensemble perturbation matrix in the model space ($\boldsymbol{X}^a$) is obtained by transforming the background ensemble as follows and used in the subsequent forecast step as initial condition:

$$\boldsymbol{X}^a = \boldsymbol{X}^b \left[(k-1)\widetilde{\boldsymbol{P}}^a\right]^{1/2} (8)$$

In the data assimilation analysis, covariance localization is applied so that the covariance among unrelated or weakly related variables is neglected. This removes the influence of spurious correlations resulting from the limited size of the ensemble. Further, it removes the influence of remote observations that may cause sampling errors. The data assimilation settings such as localization length used in this study are given Section 2.6. Estimation of emissions is based on a state augmentation technique that uses the background error correlations for each grid point to determine the relationship between the concentrations and emissions of various species (Miyazaki et al., 2012a). More detailed description of the basic data assimilation framework is available in Miyazaki et al. (2017).

## 2.2 Forecast models

We applied a common data assimilation framework to four CTM frameworks: GEOS-Chem, AGCM-CHASER, MIROC-Chem, and MIROC-Chem-H. The specifications of these systems are summarized in Table 1.

### 2.2.1 GEOS-Chem

The GEOS-Chem model is driven by assimilated meteorological data from the Goddard Earth Observing System (GEOS-5) of the NASA Global Modeling and Assimilation Office (GMAO). The adjoint model version 35 (Henze et al., 2007), which corresponds to version 9 of the forward model, with a horizontal resolution of 2° x 2.5° and 47 vertical levels extending from the surface to 0.1 hPa was used as a forward forecast model (i.e., without adjoint calculations) in this study. Although newer and improved versions of the forward model are available, we chose this version (the latest version of the adjoint model) so that an intercomparison study of 4D-VAR and EnKF using the same modelling system can be conducted in a separate study. The core of GEOS-Chem is a chemical module that includes 43 species and 318 chemical reactions and computes the local changes in atmospheric concentrations due to emissions, chemical reactions, and deposition. Further, it can simulate coupled aerosol-oxidant chemistry in the troposphere and stratosphere. This model uses the advection algorithm developed by Lin and Rood (1996) on the rectilinear grid. Convective transport is computed from the convective mass fluxes available in the



meteorological archive. The application of the EnKF chemical data assimilation system based to the GEOS-Chem model is newly developed in this manuscript.

The a priori emission data for NOx and CO were obtained from the Emission Database for Global Atmospheric Research (EDGAR) version 3 inventory (Oliver et al., 2001) for global anthropogenic emissions and from the monthly Global Fire
Emissions Database (GFED) version 2 inventory (van der Werf, 2006) for biomass burning emissions. Volatile organic compound (VOC) emission data were obtained from the RETRO inventory (Schultz et al., 2008). Emission data for North America were replaced with the 2008 National Emissions Inventory (NEI).

### 2.2.2 AGCM-CHASER

The chemical atmospheric general circulation model for the study of atmospheric environment and radiative forcing
(CHASER, Sudo et al., 2002) simulates tracer transport, wet and dry deposition, and emissions, taking into consideration of 88 chemical and 25 photolytic reactions with 47 chemical species. It has a horizontal resolution of T42 (2.8° x 2.8°) and 32 sigma levels from the surface to 4 hPa. This model is coupled to the Center for Climate System Research/National Institute for Environmental Studies (CCSR/NIES) atmospheric general circulation model (AGCM) version 5.7b. The AGCM fields in this model are nudged towards the National Centers for Environmental Prediction/Department of Energy Atmospheric
Model Intercomparison Project II (NCEP-DOE/AMIP-II) reanalyses (Kanamitsu et al., 2002) at each time step of the AGCM (i.e. every 20 min) to reproduce past meteorological conditions. The data assimilation system based on the AGCM-CHASER model (Miyazaki et al., 2012a, 2012b; Miyazaki and Eskes, 2013) was used to conduct our first chemical reanalysis calculation for 2005-2012 (TCR-1, Miyazaki et al., 2015) and elucidate the 3-D structures of lightning-induced NOx (LNOx) source (Miyazaki et al., 2014).

The anthropogenic NOx and CO emissions were obtained from the EDGAR version 4.2. Emissions from biomass burning are based on the GFED version 3.1 (van der Werf et al., 2010), while those from soils are based on the monthly Global Emissions Inventory Activity (GEIA) (Graedel et al., 1993). Using the settings reported by Lotos-Euros (Schaap et al., 2008) and Boersma et al. (2008), a diurnal variability scheme developed by Miyazaki et al. (2012a) was applied for surface NOx emissions depending on the dominant category for each area (anthropogenic, biogenic, and soil emissions). LNOx sources
were determined based on the relationship between lightning activity and cloud top height (Price and Rind, 1992) and using the convection scheme of the AGCM. Biogenic emissions from vegetation are considered for non-methane hydrocarbons (NMHCs) based on Guenther et al. (2006). Oxidations of ethane, propane, ethene, propene, isoprene, and terpenes were included explicitly.

### 2.2.3 MIROC-Chem

MIROC-Chem is the chemistry component of the MIROC-Earth system model (ESM) and is coupled to the MIROC-AGCM version 4 (Watanabe et al., 2011). It considers 92 chemical species and 262 chemical reactions (58 photolytic, 183 kinetic, and 21 heterogeneous reactions) in the troposphere and stratosphere. It has a horizontal resolution of T42 (2.8° x





2.8°) and 32 eta vertical levels from the surface to 4.4 hPa. Its tropospheric chemistry was developed based on the CHASER model with updates related to chemical reactions and emissions. MIROC-Chem considers the fundamental chemical cycle of Ox-NOx-HOx-CH$_4$-CO along with oxidation of non-methane VOCs (NMVOCs) to accurately represent ozone chemistry in the troposphere. Its stratospheric chemistry simulates chlorine and bromine containing compounds, chlorofluorocarbons

(CFCs), hydrofluorocarbons (HFCs), carbonyl sulfide (OCS), and N$_2$O. Further, it simulates the formation of polar stratospheric clouds (PSCs) and the associated heterogeneous reactions on their surfaces. The simulated meteorological fields were nudged towards the six-hourly ERA-Interim reanalysis (Dee et al., 2011). An EnKF system that is based on MIROC-Chem has been used to study decadal changes in NOx emissions (Miyazaki et al., 2017; Jiang et al., 2018). The emission data and LNOx scheme for this model are same as in the AGCM-CHASER.

**2.2.4 MIROC-Chem-H**

A high-resolution version of the MIROC-Chem model, with a horizontal resolution of T106 (1.1° x 1.1°) and 32 hybrid (eta) vertical levels, MIROC-Chem-H (Sekiya et al., 2018), was also used. This model utilizes the same chemical and transport module as the MIROC-Chem (c.f., Section 2.2.3) and has been used to study processes controlling air quality in east Asia during the KORUS-AQ aircraft campaign (Miyazaki et al., 2019; Thompson et al., 2019) and conduct the second

tropospheric chemical reanalysis (TCR-2, Miyazaki et al., in prep) for 2005-2018. Kanaya et al. (2019) demonstrated the overall good performance of the ozone and CO analyses in TCR-2 over remote oceans using observations from research vessels.

Data for anthropogenic emissions of NOx and CO were obtained from the HTAP version 2 inventory for 2010 (Janssens-Maenhout et al., 2015). This inventory combines nationally reported emissions data with data from regional scientific

inventories of the European Monitoring and Evaluation Programme (EMEP), Environmental Protection Agency (EPA), Greenhouse Gas-Air Pollution Interactions and Synergies (GAINS), and Regional Emission Inventory in Asia (REAS). Emissions from biomass burning were based on the monthly GFED version 4.2 inventory (Randerson et al., 2018) for NOx and CO, while those from soils were based on the monthly GEIA inventory (Graedel et al., 1993) for NOx. Emission data for other compounds were taken from the HTAP version 2 and GFED version 4 inventories.

As summarized in Table 1 and described in Section 2.3, the satellite products used in MIROC-Chem-H were more recent than those used in the other three models. Diversity among the data assimilation systems was enhanced by the use of different assimilated data. Although the effects of varying assimilated measurements need careful evaluation, the recently developed retrievals products reveal rather similar characteristics in general. We thus expect that the forecast model performance has a greater influence on data assimilation analysis.



## 2.3 Assimilated measurements

To assimilate satellite measurements, we have developed an observation operator ($H$) for individual assimilated measurements. This operator includes the spatial interpolation operator ($S$), a priori profile for the satellite retrievals ($\boldsymbol{x_{apriori}}$), and averaging kernel ($A$), which maps the model fields ($\boldsymbol{x_i^b}$) into the retrieval space ($\boldsymbol{y^b}$), as follows:

$$\boldsymbol{y_i^b} = H(\boldsymbol{x_i^b}) = \boldsymbol{x_{apriori}} + A(S(\boldsymbol{x_i^b}) - \boldsymbol{x_{apriori}}) \quad (9)$$

The averaging kernel captures the vertical sensitivity profiles of the retrievals (e.g., Eskes and Boersam, 2003; Jones et al, 2003; Migliorini et al, 2008). Even though the retrieval $\boldsymbol{y^o}$ and the model equivalent $\boldsymbol{y_i^b}$ depend on the a priori profile, using the averaging kernel removes the dependence of the analysis on model–retrieval comparison.

Biases in the assimilated satellite retrievals can degrade data assimilation performance. However, since the data is the same
for all comparisons with different models, the differences with respect to independent observations are relatively independent of those biases.

### 2.3.1 OMI and SCIAMACHY NO₂

The tropospheric $NO_2$ column retrievals from the DOMINO version 2 for Ozone Monitoring Instrument (OMI) and Scanning Imaging Absorption Spectrometer for Atmospheric Chartography (SCIAMACHY) (Boersma et al., 2011),
obtained from the Tropospheric Emission Monitoring Internet Service (TEMIS) website ([www.temis.nl](www.temis.nl)), were used for the GEOS-Chem, AGCM-CHASER, and MIROC-Chem systems. For MIROC-Chem-H, retrievals from the QA4ECV version 1.1 level 2 (L2) product for OMI (Boersma et al., 2017a) and SCIAMACHY (Boersma et al., 2017b) were used. Low-quality data were excluded following the published recommendations (Boersma et al., 2011; Boersma et al., 2018b).

We employed a super-observation approach to produce representative data with the horizontal resolution of each forecast
model, following the approach of Miyazaki et al. (2012a). Super-observation error was estimated using the provided retrieval uncertainty and considering an error correlation of 15 % among the individual satellite observations within a model grid cell and representativeness errors in all the systems.

### 2.3.2 TES ozone

The Tropospheric Emission Spectrometer (TES) ozone retrievals used are the version 5 level 2 nadir data obtained from the
global survey mode (Bowman et al, 2006; Herman and Kulawik, 2013) for the GEOS-Chem, AGCM-CHASER, and MIROC-Chem systems. The version 6 level 2 nadir data were used for the MIROC-Chem-H system. This data set consists of 16 daily orbits with a spatial resolution of 5–8 km along the orbit track. The standard quality flags were used to exclude low-quality data. The data assimilation of the TES ozone retrievals was performed based on the logarithm of the mixing ratio following the retrieval product specification (Bowman et al., 2006).



### 2.3.3 MLS ozone and HNO₃

The Microwave Limb Sounder (MLS) data used were the version 3.3 ozone and $HNO_3$ L2 products (Livesey et al., 2011) for all models expect MIROC-Chem-H, which used the version 4.2 data. We used MLS data for pressures of less than 215 hPa for ozone and less than 150 hPa for $HNO_3$, while tropical-cloud-induced outliers were excluded. The provided accuracy and precision of the measurement error were included as the diagonal element of the observation error covariance matrix.

### 2.3.4 MOPITT CO

The version 6 level 2 thermal infrared (TIR) products (Deeter et al., 2013) of the Measurement of Pollution in the Troposphere (MOPITT) were used for all models except the MIROC-Chem-H, for which the version 7 level 2 TIR/near infrared (NIR) total column CO data were used (Deeter et al., 2017). Owing to data quality problems, we excluded data poleward of 65° and night-time data. For the version 6 TIR products, data at 700 hPa were used for constraining surface CO emissions. For the version 7 TIR/NIR products, the total column averaging kernel was used in the observation operator to estimate simulated total columns. The uncertainty information provided in the retrievals was used in the observation error. Like in the case of $NO_2$ measurements, the super-observation approach was applied for MOPITT measurements as well.

### 2.4 Validation data

### 2.4.1 WOUDC ozonesonde data

All available ozonesonde observations taken from the World Ozone and Ultraviolet Radiation Data Center (WOUDC) database (available at http://www.woudc.org) were used as validation data. All ozonesonde profiles have been interpolated to a common vertical pressure grid, with a bin of 25 hPa. The ozone fields from the control and data assimilation calculations were linearly interpolated to the time and location of each measurement, with a bin of 25 hPa, and then compared with the measurements at 4° x 4° grid points.

### 2.4.3 WDCGG surface carbon monoxide

Surface CO concentration observations were obtained from the World Data Centre for Greenhouse Gases (WDCGG) operated by the World Meteorological Organization (WMO) Global Atmospheric Watch programme (http://ds.data.jma. go.jp/gmd/wdcgg/). Hourly and event observations from 59 stations were used to validate surface CO concentrations from the control and data assimilation runs at 5° x 5° grid points.



## 2.5 Multi-model analysis

We construct integrated data assimilation analysis using multiple models combined with multiple-species measurements. The multi-model integrated analysis $\overline{x_m}$ are obtained by combining data assimilation analyses ($x_j^a$) weighted by analysis uncertainties ($\sigma_j^2$) of individual models (j=1–4) as follows:

$$\overline{x_m} = \frac{\sum\left(x_j^a / \sigma_j^2\right)}{\sum\left(1/\sigma_j^2\right)} \quad (10)$$

The integrated analysis ($\overline{x_m}$) provide unique information on atmospheric states, which are less dependent on the characteristics of individual models used for data assimilation, and consider the uncertainty of individual data assimilation analyses. The uncertainty of the integrated analysis ($\overline{\sigma_m^2}$) is defined as follow:

$$\overline{\sigma_m^2} = \frac{1}{\sum\left(1/\sigma_j^2\right)} \quad (11)$$

We apply this approach for estimating multi-model mean ozone fields in this study. Because of the predefined minimum values of the standard deviations applied to surface emissions of emissions of CO and $NO_2$ to prevent covariance underestimation during data assimilation (c.f., Section 2.6), the analysis spreads of near surface NOx and CO concentrations are not fully meaningful. For the concentrations and emissions of CO and NOx, therefore, its multi-model mean and uncertainty were estimated as a standard ensemble mean and spread, without using the analysis uncertainty of individual models.

## 2.6 Experimental setting

We conducted one year of data assimilation calculations and forward model simulations (i.e., control run) from January 1, 2007, with a two-month spin up from November 1, 2006, using the four systems. This assimilation period was chosen to provide comprehensive constraints by OMI measurements while avoiding the influences of OMI row anomalies (December 2009 onwards) (Schenkeveld et al., 2017) and reduced numbers of the TES measurements (2010 onwards). A control run was performed in each system using the same model settings as the data assimilation run but without performing data assimilation. The validation results for the control and data assimilation runs were compared to measure the improvements achieved through data assimilation in each system.

The data assimilation settings were almost same among the systems as follows. The state vector includes the chemical concentrations of various species as well as the surface sources of NOx and CO and LNOx sources. The LNOx source optimization is based on the scheme developed by Miyazaki et al. (2014). For the MIROC-Chem-H system, the state vector also includes surface $SO_2$ emissions, as implemented in Miyazaki et al. (2019). The state vectors for the MIROC-Chem and MIROC-Chem-H systems include a correction factor for emission diurnal variability to improve the representation of diurnal emission variability using the OMI and SCIAMACHY retrievals obtained at different overpass time, based on the scheme developed by Miyazaki et al. (2017).



Covariance inflation was applied to analyses of both concentrations and emissions to prevent underestimation of background error covariance and filter divergence caused by sampling errors associated with the limited ensemble size and by model errors, following the settings used by Miyazaki et al. (2015). Further, localization was applied to avoid the influence of remote observations that may cause sampling errors, with a cut-off radius of approximately 1650 km for NOx

emissions and 2000 km for CO emissions, LNOx sources, and chemical concentrations, as in Miyazaki et al. (2015). We also applied covariance localization for different variables in the state vector (Kang et al., 2011), by setting the covariance among non- or weakly related variables to be zero. The analysis of surface emissions of NOx and CO allowed for error correlations with $NO_2$ and CO concentrations only, respectively. For LNOx sources, covariances with CO data were neglected. Assimilation of MOPITT CO data was used to constrain surface CO emissions only.

The a priori error was set to 40% for surface emissions of NOx and CO and 60 % for LNOx sources, which are comparable to the reported emission uncertainty (e.g., Schumann and Huntrieser, 2007; Kaiser et al., 2012; Li et al., 2017). To prevent covariance underestimation and maintain emission variability during the long-term assimilation calculation, we applied covariance inflation to the emission source factors in the analysis step. The standard deviation of the emission source factors was artificially inflated to a minimum predefined value (30% of the initial standard deviation) at each analysis step.

The data assimilation cycle was set to be two hours for the AGCM-CHASER, MIROC-Chem, and MIROC-Chem-H systems, and six hours for the GEOS-Chem system because of the limitation associated with meteorological data input in GEOS-Chem. The emission and concentration fields were analyzed and updated at each analysis step in all the systems. We have confirmed that the results of data assimilation can differ when the data assimilation cycle is changed from two hours to six hours using the AGCM-CHASER system. This occurs, in particular, for the analysis of short-lived species with strong

diurnal variability and NOx emission estimates. The performance of the GEOS-Chem data assimilation can thus be expected to differ with the use of a two-hour data assimilation cycle and meteorological data inputs with higher temporal frequency for short-lived species.

In summary, there are differences in the assimilated measurements (updated retrievals were used in MIROC-Chem-H), diurnal emission variability (data assimilation corrections were made in the MIROC-Chem and MIROC-Chem-H systems

only), and data assimilation cycle (six hours in GEOS-Chem) of the four systems. These differences will lead to discrepancies in the data assimilation analyses of the four systems attributable to assimilation system configuration rather than the forward models themselves. While impact of these configurations can be further refined in future studies, the major discrepancies in the data assimilation analyses are still primarily attributable to the models themselves.

## 3 Data assimilation statistics

### 3.1 Analysis increment

The analysis increment (Eq. 7) information is a measure of the adjustment made in the analysis step, which is estimated from the differences between the forecast and the analysis after each analysis step. As shown in Fig 1a, the annual mean



analysis increments are largely different among the models, reflecting different systematic model biases. For individual systems, the analysis increments are in good agreements with the OmF (Eq. (6)). This confirms that the model errors were effectively reduced using data assimilation.

In the ozone concentration field at 500 hPa, the AGCM-CHASER system gives large positive increments in the extratropics

of both hemispheres, with annual mean values in the range of 1–3 ppb/day, whereas the increments are negative at low latitudes (up to -1.5 ppbv/day). The standard deviations of the analysis increment are 0.8–1.7 ppb/day in the extratropics and 0.2–0.4 ppb/day at low latitudes. The analysis increments are relatively low in GEOS-Chem (up to -1.8 ppbv/day) and MIROC-Chem (up to 1.4 ppbv/day) in the NH extratropics; in GEOS-Chem (-0.5–1.5 ppbv/day) and MIROC-Chem-H (up to -1.0 ppbv/day) in the tropics; and in MIROC-Chem (up to 1.4 ppbv/day) in the SH extratropics. GEOS-Chem exhibits

negative increments except over central Africa and northern South America, with large negative increments (up to 2 ppbv/day) over the Southern Ocean and the U.S. west coast in the strong westerlies and Aleutian Low regions, respectively. The positive increments over central Africa and northern South America could imply underestimated ozone productions due to biomass burning or VOCs emissions.

The analysis increments differed significantly between the lower and upper troposphere as well as among seasons in all the

systems (figure not shown). GEOS-Chem shows large positive increments (0.5–2.2 ppbv/day) in the extratropics at 700 hPa, in contrast to negative increments (up to -2.0 ppbv/day) at low and mid latitudes at 350 hPa. In AGCM-CHASER and MIROC-Chem, the increments changed from positive at 700 hPa (up to 2.2 ppb/day in AGCM-CHASER and 0.5 ppb/day in MIROC-Chem) to negative at 350 hPa (up to -2.5 ppb/day and -1.2 ppb/day, respectively) in the extratropics of both hemispheres. The positive increments in MIROC-Chem-H decreased with height in the extratropical troposphere. As the

increments in the troposphere are mainly introduced by the TES assimilation, the vertical structures suggest that the assimilated TES ozone measurements have independent information regarding the lower- and upper-tropospheric ozone. Jourdain et al. (2007) showed that the TES retrievals have 1–2 DOFs in the troposphere, with the highest number of DOFs for the clear-sky tropics and subtropics. The seasonal changes in the analysis increment reflect variations in the short-term systematic model errors and observational constraints, which also differed significantly among the models.

**3.2 Analysis uncertainty**

The analysis uncertainty, which is estimated as the standard deviation of the analyzed concentrations across the ensemble (Eq. 8) in individual systems, can be used as a measure of the uncertainty of each data assimilation analysis. The analysis uncertainty is due to errors in the model input data, model processes, and assimilated measurements and is reduced as the analyses converges to the true state. Because the model input data and assimilated measurements are almost same among the

models, differences in model processes such as response of ozone to perturbed emissions and chemical lifetimes should be primarily responsible for the analysis spreads among the models through the forecast step. Detailed investigation on the impact of different model processes for each region and season would be helpful to interpret the results but is beyond the



scope of this paper. The simultaneous emissions and concentration optimization were important in producing appropriate ensemble perturbations in ozone, especially in the lower and middle troposphere.

The ozone analysis uncertainty at 500 hPa shown in Fig. 1b is generally smaller in the tropics than in the extratropics likely a consequence of the higher sensitivities in the TES ozone retrievals in the tropics. Because common settings were applied to the ensemble size and covariance inflation, the obtained inter-model differences in the spread reflect different systematic model errors related to the assimilation window size. The annual mean analysis uncertainty is generally larger in AGCM-CHASER and MIROC-Chem than in GEOS-Chem and MIROC-Chem-H. In the tropics, the analysis uncertainty is approximately 2–5 ppb in GEOS-Chem and MIROC-Chem-H and approximately 5–11 ppb in AGCM-CHASER and MIROC-Chem. In the extratropics, the analysis spread is approximately 6–10 ppb in GEOS-Chem and MIROC-Chem-H and 10–16 ppb in AGCM-CHASER and MIROC-Chem. The analysis increments are generally similar among the models (c.f., Fig. 1a). These results suggest that the model forecasts tended to diverge more quickly in AGCM-CHASER and MIROC-Chem, likely as a result of larger differences in the equilibrium state between the model and assimilation. In the UTLS region, the analysis uncertainty is relatively smaller in the extratropics than in the tropics, because of the high accuracy of the MLS measurements. The spatial pattern in GEOS-Chem and MIROC-Chem-H are remarkably similar whereas the CHASER and MIROC patterns are much more similar.

The multi-model standard deviation of the ozone analyses (typically <5 ppb for the globe, Fig. 2c) is significantly lower than the analysis uncertainty in AGCM-CHASER and MIROC-Chem (Fig. 1b). As will be discussed in Section 4,1, mean errors against independent observations are also significantly smaller than the analysis uncertainty in these models. These results indicate that the analysis uncertainty depends on the choice of forward model and was possibly overestimated in AGCM-CHASER and MIROC-Chem because of a large diversity in forecast trajectories. The overestimated analysis error covariance was also confirmed by smaller chi squares (e.g., Ménard, R. and Chang, 2000) in these models (not shown). To measure the analysis spread corresponding to the actual analysis uncertainties, additional observational information and optimizing the covariance inflation to the forecast error covariance would be required.

### 3.3 Multi-model integrations

Fig 2a shows the integrated ozone analysis fields, $\overline{x_m}$ defined in Eq. (10), that were created using MOMO-Chem. The annual and multi-model mean ozone concentrations at 500 hPa are high in the NH extratropics (55–70 ppbv) and lowover the Maritime continent and the tropical western Pacific (22–35 ppbv). Because the analyses from the GEOS-Chem and MIROC-Chem-H systems exhibit smaller analysis spreads (c.f., Section 3.3.2), they exert a strong control on the integrated fields. At 500 hPa, the estimated uncertainty of the integrated fields, $\overline{\sigma_m^2}$ defined in Eq. (11), is 2–4.5 ppbv in the NH, 0.5–2 ppbv in the tropics, and 3–5.5 pp bv in the SH (Fig. 2b). These values are smaller than the uncertainties of the individual model analyses (Fig. 1b), demonstrating that the integrated fields can provide more reliable and unique information. The multi-model spread of individual data assimilation analysis (Fig. 2c) is typically smaller than the multi-model mean integrated uncertainty (Fig. 2b). Again, with the multi-model spread (Fig. 2c) and the differences with the ozonesonde measurements



(Section 4.1) being smaller than the multi-model mean uncertainty (Fig. 2b), the comparisons suggest that the analysis uncertainty might be overestimated in some of the analyses.

Over the northern South America, the larger multi-model spread compared to the multi-model mean uncertainty suggests that the background errors might have been underestimated, as rapid error growths due to deep convection and biomass
burning might have not been accounted for properly. Differences in isoprene emissions and chemistry could also enhance the multi-model spread over the region (Archibald et al., 2010). Techniques such as adaptive inflation for background error covariance (e.g., Anderson, 2007) would be helpful to represent rapid changes in background errors in the individual models.

### 3.4 Ozone and NO₂ response to NOx emissions

### 3.4.1 Multi-model comparisons

From a system analysis perspective, one of the fundamental questions in atmospheric chemistry is the sensitivity of a constituent like ozone to changes in surface emissions such as NOx emissions. With recent advances in estimating preindustrial ozone (Yeung et al, 2019), model sensitivities are the primary drivers of chemistry-climate estimates of quantities such as ozone radiative forcing (Bowman et al, 2013, Myhre et al., 2013). While these simulations describe relatively slow, equilibrium responses, data assimilation incremental updates provide statistics on "fast" responses within the
short data assimilation windows. By simultaneously updating ozone and NOx emissions, multi-constituent data assimilation can yield insight into this fundamental quantity. We explore this potential by regressing both the ozone and NO₂ increments with respect to the NOx emission analysis increments.

As summarized in Table 2, the response of ozone and NO₂ analysis to emission perturbations (i.e., data assimilation increments) is largely different among the models. The NO₂ surface response to NOx emissions is well-correlated
(correlation >= 0.93 for all models) but the response differs by almost a factor of 2 between GEOS-Chem and MIROC-Chem-H. Globally, this diversity holds between surface ozone concentration and NOx emission increments ($\frac{\Delta O_3}{\Delta ENOx}$) for these two models. However, the AGCM-CHASER ozone-NOx emissions response (1.5 ppb/($10^{-11}$ kgNm$^{-2}$s$^{-1}$)) is the largest among all the models. On the other hand, the correlation between surface ozone and NOx emissions is relatively weak (correlation < 0.43) reflecting the much more complicated chemical and dynamical relationship. For polluted areas (greater than $3\times10^{-11}$
kgNm$^{-2}$s$^{-1}$, as shown in the brackets in Table 2), the largest response is AGCM-CHASER and MIROC-Chem, which is greater than the other two models by 40–180 %, with similar intercepts and correlations. The multi-model diversity reflects the different representation of NOx and VOC as well as dynamics, leading to different ozone production efficiencies. In the case of GEOS-Chem and MIROC-Chem-H, there appears to be a clearer relationship between $\frac{\Delta NO_2}{\Delta ENOx}$ and $\frac{\Delta O_3}{\Delta ENOx}$, suggesting that NOx chemistry plays a more dominant role in ozone formation than other factors. By separating these two responses,
MOMO-Chem is able to quantify the responses of forward models with unique diagnostics, without making any sensitivity calculations.



Different model responses would directly impact the Kalman gain in Eq. 4, leading to a more efficient model error reduction. Given the same predefined minimum values for the surface NOx emission perturbation (c.f., Section 2.4), a larger ozone analysis uncertainty (through a larger forecast model spread) would be obtained in models with a stronger ozone response to NOx emissions. In fact, stronger ozone response (Table 2) and larger analysis uncertainty (Fig. 1b) are consistently found in

AGCM-CHASER and MIROC-Chem. Meanwhile, the ozone response to a given perturbation is dependent on the background condition because of the non-linear $O_3$-NOx chemistry (e.g., Zaveri et al., 2003). The multi-constituent framework allows to evaluate model ozone response in a realistic condition while considering possible error ranges in precursor's emissions using emissions analysis increments (c.f., Section 5). The ozone analysis increments became substantially smaller in all the models for most cases by including the emission optimization, and the increments could be

regarded as inherent and persistent model biases of individual models. Therefore, a systematic investigation of model ozone response and analysis increment in the multi-constituent data assimilation framework could benefit evaluation of future prediction of chemistry-climate system as a hierarchical emergent constraint (Bowman et al., 2018) and for making effective ozone control strategies.

### 3.4.2 Implications for chemistry model predictions

By applying linear regressions to the multi-model integrated fields (c.f., Section 3.3), we evaluated model responses of surface ozone and $NO_2$ concentrations to NOx emissions. As shown by Fig. 3, the estimated model responses from the MOMO-Chem integrated fields provide unique information on fast responses to NOx emissions. The surface $NO_2$ response exhibits a large seasonal variation in the NH, with a maximum value of about 2 ppb/($10^{-11}$ kgNm$^{-2}$s$^{-1}$) in January, reflecting the longer chemical lifetime of NOx in winter. The rapid increases from September to December and decreases from January

to March can be associated primarily with variations in temperature, OH, and $NO_2$ photolysis. The annual mean slope is about 40 % smaller in the tropics than in the NH (0.70 vs. 1.15 ppb/($10^{-11}$ kgNm$^{-2}$s$^{-1}$)), because of the shorter chemical lifetime of NOx in the tropics. The surface $NO_2$ and NOx emissions in the integrated fields are well-correlated (coefficients>0.9) throughout the year both in the NH and the tropics. The inter-model differences (red shade) increase in winter in the NH, with a maximum standard deviation of 35 % in January, implying strong model-dependence of surface

$NO_2$ given same NOx emissions.

The ozone response shows an opposite seasonal cycle to the $NO_2$ response in the NH. It gradually increased from January to August by about 0.4 ppb/($10^{-11}$ kgNm$^{-2}$s$^{-1}$)/month. It reaches 2.4 ppb/($10^{-11}$ kgNm$^{-2}$s$^{-1}$) in August with relatively large coefficients (0.3–0.6) in May–September. The large ozone response implies substantial photochemical productions of surface ozone over polluted areas in summer. Then, the slope decreases rapidly from August to October by about 1.1

ppb/($10^{-11}$ kgNm$^{-2}$s$^{-1}$)/month, and it becomes negative in winter but with low coefficients (-0.3–0.2). The negative slopes, with a minimum value of -0.6 ppb/($10^{-11}$ kgNm$^{-2}$s$^{-1}$) in January, could be driven by the dilution effects over highly polluted areas.





In the tropics, the ozone response is stronger than in the NH (3.1 vs. 0.6 ppb/($10^{-11}$ kgNm$^{-2}$s$^{-1}$) for annual mean), with strongest responses of about 4.3 ppb/($10^{-11}$ kgNm$^{-2}$s$^{-1}$) in March and October. The different ozone production efficiency implies that any latitudinal shifts in NOx emissions from the extratropics to the tropics would lead to increases in global tropospheric ozone, as suggested by Zhang et al. (2016), while showing strong seasonality. Our analysis indicates that the

mean ozone response is comparable between the NH and the tropics in August and September. The seasonal variation in the tropics is likely associated with biomass burning events (e.g., Bowman et al., 2009; Jones et al., 2009; Parrington et al., 2012), with enhanced ozone responses over Southeast Asia during February–June (2.3–3.7 ppb/($10^{-11}$ kgNm$^{-2}$s$^{-1}$)), over central America and tropical South America during April–July (2.8–5.3 ppb/($10^{-11}$ kgNm$^{-2}$s$^{-1}$)), over central Africa in March (5.5 ppb/($10^{-11}$ kgNm$^{-2}$s$^{-1}$)) and October (7.1 ppb/($10^{-11}$ kgNm$^{-2}$s$^{-1}$)), and over India in March and October (5.1 ppb/($10^{-11}$

kgNm$^{-2}$s$^{-1}$)). Although the surface ozone and NOx emissions are well correlated in the multi-model integrated analysis throughout the year (coefficients>0.5), the large multi-model spreads (25–55%) suggest that individual models have large uncertainty in representing strong ozone productions, for instance, associated with VOCs emissions and chemistry. The inter-model correlations of $\Delta ENOx$ and $\Delta O_3$ were strongly dependent on season and location (not shown), which also provide information on the robustness (i.e., multi-model diversity) of the estimated ozone and NO$_2$ responses for each

location and season.

Finally, the model responses differ significantly between the MOMO-Chem integrated fields (solid blue and red lines) and the mean of the individual model estimates (solid while lines), especially when the model responses are strong. The multi-model integrated fields exhibit about 20 % larger NO$_2$ response in December and about 70 % larger ozone response in August than the mean of the individual model estimates in the NH. In the tropics the monthly ozone response is up to about

60 % larger in the multi-model integrated analysis. The different responses reflect non-Gaussian distributions of the individual model fields. The results imply that the observationally constrained, multi-model integrated fields provide fundamentally different fast chemical processes than those in the individual models. With further investigations of the chemical relationships in the integrated fields, the MOMO-Chem framework would provide insights into ozone production processes to inform chemical predictions through relationships such as emergent constraints (Bowman et al., 2018). This

example demonstrates the unique capability of the MOMO-Chem framework for various applications.

## 4. Validation results

### 4.1 Ozone profiles

#### 4.1.1 Comparisons against TES observations

Fig. 4 compares the annual zonal mean ozone from the lower to upper troposphere. In comparison with the TES

measurements, at 750 hPa, all the control runs underestimate the mean ozone in the NH extratropics (by -4.4 to -3.2 ppb at 50°N). At low latitudes, the mean ozone in MIROC-Chem-H is underestimated by -6 to -3 ppbv. In the SH extratropics, all



the models reproduced the lower tropospheric ozone well. At 510 hPa, the zonal mean biases differ obviously among the models, with multi-model standard deviations of 1.5–4 ppv in the SH, 3.2–5 ppb in the tropics, and 3–6.6 ppb in the NH. The biases are largely negative in GEOS-Chem (-7.4 ppb at 50°N) and AGCM-CHASER (-5.3 ppb) in the NH extratropics; they are negative in MIROC-Chem-H (-11 to -6 ppb) at low latitudes, and positive in the models except GEOS-Chem (3.2 to 5.1 ppb at 50°S) in the SH extratropics. Similarly, at 316 hPa, the biases obtained using the models are quite different, with large positive biases in MIROC-Chem and MIROC-Chem-H in the extratropics of both hemispheres and large negative biases in MIROC-Chem-H in the tropics. Global total budgets and the production rates of tropospheric ozone can also differ, as suggested by multi-model inter-comparison studies including GEOS-Chem and MIROC-Chem (Young et al., 2013; 2018; Hu et al., 2017). Sekiya et al (2018) demonstrated that the ozone chemical productions are smaller in MIROC-Chem-H (4647 Tgyr$^{-1}$ for 2008) than in MIROC-Chem (4809 Tgyr$^{-1}$).

After the data assimilation, all the models are in good agreement with the assimilated TES measurements as expected, and demonstrate improved inter-model consistency. In the NH, the mean bias at 750 hPa is reduced by 19–73% to between -4.1 and -0.4 ppb (at 50°N) in all the models. At 510hPa, the large negative model biases in GEOS-Chem and AGCM-CHASER are reduced by 76% and 92% at 50°N, respectively. In the SH, most of the large model biases in MIROC-Chem-H are removed throughout the troposphere.

Fig. 5 shows the spatial distributions of the annual mean ozone concentrations at 510 hPa. The general structure of tropospheric ozone is well reproduced by the control runs, such as the low ozone concentrations over the tropical western Pacific and the high over the Middle East. The annual and zonal mean model biases are negative in the tropics in all the models, with large negative biases over the southern Atlantic; the bias is largest in MIROC-Chem-H (by up to 20 ppbv). After data assimilation, most of the model biases are removed for the globe. In the extratropical UTLS (figure not shown), the remaining mean bias was close to the mean observational error of the MLS ozone measurements in all the systems.

As shown in Fig. 6a, the multi-model standard deviation of the annual mean ozone at 510 hPa obtained from the control runs, with applying the TES averaging kernels (AKs), are typically 5–10 ppb from the tropics to the NH high latitudes and 1–5 ppb in the SH extratropics. After the data assimilation, the standard deviation mostly becomes smaller than 5 ppb and 3 ppb for these regions, respectively, with reductions for the zonal mean values by 20–60% in the NH and 30–85% in the SH. The results demonstrate that the assimilation framework provides highly consistent analysis fields among the systems, less dependent on the performance of the individual models. The obtained multi-model standard deviation after data assimilation (Fig. 6b) is comparable to the mean model errors relative to the TES measurements for most regions, which could thus be used as an estimate of the mean data assimilation uncertainty.

**4.1.2 Comparisons against ozonesonde observations**

The current ozonesonde network is heterogeneously distributed globally with a sampling interval typically a week or longer. Model errors are also expected to vary greatly in time and space at various scales. As a consequence, the ozonesonde measurements suffer from significant sampling bias. Miyazaki and Bowman (2017) demonstrated that this ozonesonde



sampling bias in the evaluated model bias for the seasonal mean concentration relative to global coverage reaches 80 % for the global tropics. Nevertheless, the ozonesonde network provides a critical independent validation of the data assimilation products, while the data assimilation products are advantageous for evaluating actual regionally and seasonally representative model performance, which are required for model improvements. The synergy of the two provides a

mechanism to characterize chemical reanalysis evaluation of chemistry-climate models (Miyazaki and Bowman, 2017).

Fig. 7 compares the seasonal variation of ozone with the WOUDC global ozonesonde measurements from the lower troposphere to the lower stratosphere. In the lower troposphere (850–500 hPa), all the models mostly underestimate ozone at NH mid and high latitudes, except for GEOS-Chem at NH mid latitudes in boreal summer. The negative model biases are large at NH high latitudes in boreal spring, with an annual mean bias of -4.7 to -2.6 ppbv (as summarized in Table 3) and

large multi-model spreads. In the tropical lower troposphere, the models, other than MIROC-Chem-H, mostly overestimate ozone except in September–October, whereas MIROC-Chem-H underestimates the annual mean ozone by 5.8 ppbv. In the SH, all the models underestimate ozone throughout the year, with an annual mean bias of -6.2 to -0.7 ppbv at mid latitudes and -4.6 to -2.2 ppbv at high latitudes. The negative model biases in the SH have been found in most of the chemistry-climate models in the ACCMIP project (Bowman et al., 2013; Young et al., 2013).

In the middle and upper troposphere (500–200 hPa), the model biases reveal a large diversity at NH high latitudes. The enhanced multi-model spread in spring could be associated with the different representations of the stratosphere-troposphere exchange (STE) processes. At NH mid latitudes, MIROC-Chem and MIROC-Chem-H overestimate annual mean ozone by 16.1 and 4.1 ppbv, respectively. In the tropics, the models, other than MIROC-Chem-H, overestimate ozone in boreal winter and underestimate it in boreal autumn, thus underestimating the seasonal amplitudes. In the SH, all the models overestimate

ozone with an annual mean bias of 2.8–20.5 ppbv at mid latitudes and 8.7–29.9 ppbv at high latitudes. In the upper troposphere and lower stratosphere (UTLS, 200–80 hPa), the large multi-model spread can primarily be due to the different representations of the stratospheric chemistry, STE, as well as convective transport in the tropics. Large positive model biases exist in MIROC-Chem and MIROC-Chem-H in the NH extratropics, MIROC-Chem and GEOS-Chem in the tropics, and all the models in the SH extratropics.

Because of data assimilation, the large negative model biases in the lower troposphere are largely reduced in the NH lower troposphere in boreal spring. Nevertheless, the annual mean concentrations in all the systems become too high in the NH lower troposphere, with an annual mean bias of from 1.7 to 4.3 ppb at high latitudes and from 1.9 to 5.2 ppb at mid latitudes, while the underestimation in the seasonal amplitude is reduced in all the models. The weak sensitivity of the assimilated measurements and the changes made to the precursor emissions (c.f., Section 5) could be responsible for the overestimations.

In the tropics, the negative model bias in boreal autumn is reduced via data assimilation, thus enhancing the seasonal amplitudes in all the system, whereas the analyzed concentrations become too high in AGCM-CHASER and MIROC-Chem-H in boreal summer. In the SH, the data assimilation reduced the negative model biases of MIROC-Chem-H at mid latitudes (from -6.2 ppb to 1.0 ppbv annual mean bias) and MIROC-Chem and MIROC-Chem-H at high latitudes (from -4.6 to -4.5 ppbv to 0.9 to 3.6 ppbv). The observed rapid increases during August–October at SH mid latitudes are well reproduced after



data assimilation in all the systems. At high latitudes of both hemispheres, some of the models exhibit too high concentrations after data assimilation. An inaccurate balance between the mid and high latitude in model transport and the lack of direct observational constraints could limit the effectiveness of data assimilation at high latitudes. Conducting observational impact analysis would help suggesting a framework to obtain a better global tropospheric ozone analysis.

Both the agreements against the observation and the multi-model consistency are greatly improved via data assimilation from the middle troposphere to the lower stratosphere for the globe, with annual mean bias reductions from -29.9 to 29.7 ppbv to -9.2 to -6.2 ppbv (i.e., by 53–81%) at NH high latitudes, from -4.0 to 16.1 ppbv to -2.5 to -0.2 ppbv (by 39–76% except for AGCM-CHASER) at NH mid latitudes, from -11.8 to 1.3 ppbv to -0.3 to 3.0 ppbv (by 50–91%) in the tropics, from 2.8 to 20.5 ppbv to -1.9 to 3.0 ppbv (by 71–94%) at SH mid latitudes, and from 8.7 to 29.9 ppbv to 1.0 to 4.7 ppbv (by

46–97%) at SH high latitudes for 500–200 hPa. The estimated mean errors are significantly smaller than the analysis uncertainty (Fig. 1b) in AGCM-CHASER and MIROC-Chem (10–16 ppb) and is comparable to that in GEOS-Chem and MIROC-Chem-H. These results suggest overestimated analysis uncertainty in AGCM-CHASER and MIROC-Chem.

Fig. 8 shows that the data assimilation introduces similar changes to the seasonal amplitudes of ozone (defined as the difference between the maximum and minimum concentrations) in the four models, such as the increases in the lower and

middle troposphere and the decreases in the extratropical upper troposphere and lower stratosphere. Between 850 and 500 hPa, the control runs underestimated the seasonal amplitude in the extratropics of both hemispheres compared with the ozonesonde measurements (e.g., by up to -29 % at the NH mid latitudes). The model underestimates are largely reduced by data assimilation in all the models. Between 500 and 200 hPa, data assimilattion mostly removed the negative bias in GEOS-Chem (-8 %) and AGCM-CHASER (-5 %) and the positive bias of the seasonal amplitude in MIROC-Chem-H (47 %)

against the ozonesonde measurements in the NH, and the large positive bias in MIROC-Chem (22%) in the SH. Between 200 and 90 hPa, positive biases are reduced in all the models globally. In the NH, the range in the bias from 13 to 40% is reduced to a range from -12 to 3%, with the largest reduction observed in MIROC-Chem-H (from 40 to 2 %). In the tropics, the range in the bias is reduced from 20 to 148% to from 10 and 25%, with the largest reduction observed in GEOS-Chem (from 148 to 10%). In the SH, the range in bias is reduced from 15 to 92% to from -1 and 19%, with the largest reduction

observed in MIROC-Chem (from 92 to 10%).

### 4.2 Tropospheric NO$_2$ columns

For the comparisons with the OMI NO$_2$ retrievals, the OMI NO$_2$ AKs from the DOMINO2 products were applied to GEOS-Chem, AGCM-CHASER, and MIROC-Chem, whereas those from QA4ECV were applied to MIROC-Chem-H, corresponding to the assimilated measurements for each system. In Fig. 9, the converted tropospheric NO$_2$ columns from the

control and assimilation runs are then compared with the assimilated OMI retrievals: the DOMINO2 product for GEOS-Chem, AGCM-CHASER, and MIROC-Chem (black line vs blue, red, and green lines in Fig. 9) and the QA4ECV product for MIROC-Chem-H (gray line vs yellow line).



As summarized in Table 4, the model bias in tropospheric $NO_2$ column differed largely among the models, because of the different model configurations (e.g., chemical lifetime of NOx) and input data (e.g., NOx emissions). The models, other than GEOS-Chem, mostly underestimate tropospheric $NO_2$ columns over polluted areas, same as in most other CTMs (van Noije et al., 2006), with an annual mean bias ranging from -2.07 to -0.37 × $10^{15}$ molecules $cm^{-2}$ over eastern China, -0.51 to -0.26 ×

$10^{15}$ molecules $cm^{-2}$ over the United States, and -0.82 to -0.32 × $10^{15}$ molecules $cm^{-2}$ over Europe. GEOS-Chem overestimates tropospheric $NO_2$ columns over some parts of China (with annual and regional mean bias of 0.13 × $10^{15}$ molecules $cm^{-2}$ over eastern China), Europe (0.60 × $10^{15}$ molecules $cm^{-2}$), and the United States (0.29 × $10^{15}$ molecules $cm^{-2}$). The model biases in tropospheric $NO_2$ columns can vary with changing the model configurations. For instance, important NOx sink pathway determining $NO_2$ simulation uncertainties include the $NO_2$+OH reaction and the formation of $HNO_3$ in

the NO+$HO_2$ reaction (Lin et al., 2012; Stavrakou et al., 2013), which are represented differently in the models. The columns simulated from MIROC-Chem-H are higher than that from MIROC-Chem, with the same AKs applied over some parts of the polluted areas such as eastern China; these differences are attributable to the increased model resolution, which suppresses the dilution effects (Sekiya et al, 2018).

Fig. 9 compares the seasonal variation of tropospheric $NO_2$. The models, other than GEOS-Chem, underestimate

tropospheric $NO_2$ columns throughout the year over eastern China, the United States, and Europe, with the largest negative biases in boreal winter. Over India, GEOS-Chem reproduced the peak observed in April and the rapid decrease from May to July, whereas the other models underpredicted the seasonal variations. The retrieved tropospheric $NO_2$ columns are generally lower in the QA4ECV products than in the DOMINO-2 products over most of major polluted areas. The different retrieved columns can largely be attributed to differences in the a priori profiles and does not influence directly the model-observation

differences after applying the AKs (Boersma et al. 2018a). Over Southeast Asia, the models, except for GEOS-Chem, underestimate the peak observed in March, which is associated with biomass burning, by 11–55%, whereas the models overestimate the peak over South America in September by 18–31%.

Over northern, central, and southern Africa, all the models underestimate tropospheric $NO_2$ columns throughout the year, with an annual mean bias ranging from -0.32 to -0.11 × $10^{15}$ molecules $cm^{-2}$, -0.39 to -0.11 × $10^{15}$ molecules $cm^{-2}$, and -1.05

to -0.89 × $10^{15}$ molecules $cm^{-2}$, respectively. Over southern Africa, the negative model bias is maximized in austral winter (by 43–63 %), with MIROC-Chem-H giving the smallest bias. The higher spatial resolution of MIROC-Chem-H is considered essential in resolving individual polluted areas in the Highveld region and in accurately simulating the non-linear effects on $NO_2$ loss rate.

The tropospheric $NO_2$ column retrievals from OMI and SCIAMACHY were assimilated to optimize NOx emissions, and

the assimilation of non-$NO_2$ measurements influence the chemical lifetime of NOx through changes made to OH. Data assimilation reduced the negative model biases in the models, other than GEOS-Chem, over eastern China (from -2.07–-0.37 × $10^{15}$ molecules $cm^{-2}$ to -0.69–-0.39 × $10^{15}$ molecules $cm^{-2}$) and the United States (from -0.51 to -0.26 × $10^{15}$ molecules $cm^{-2}$ to -0.18 to -0.13 × $10^{15}$ molecules $cm^{-2}$), and over western Europe (from -0.82 to -0.32 × $10^{15}$ molecules $cm^{-2}$ to -0.41 to -


$0.24 \times 10^{15}$ molecules cm$^{-2}$). The annual mean positive model biases in GEOS-Chem are reduced by 72% over the United States and by 65% over Europe. The temporal correlations are also improved in all the models.

Over India, the data assimilation increases tropospheric NO$_2$ columns in boreal winter-spring and reproduced the observed local maximum in May and minimum in July in all the models. Consequently, the seasonal amplitude is enhanced, leading to

improved temporal correlations (from 0.20–0.87 to 0.94–0.99) while reducing the annual mean bias (from -0.27 to -0.14×10$^{15}$ molecules cm$^{-2}$ to 0.12 to -0.02×10$^{15}$ molecules cm$^{-2}$, by 40–86%). Over Southeast Asia, the persistent model negative biases are reduced (from -0.56 to -0.15 × 10$^{15}$ molecules cm$^{-2}$ to -0.18 to 0.05 × 10$^{15}$ molecules cm$^{-2}$, by 40–77%) with improved temporal correlations (from 0.86–0.96 to 0.97–1.00) in all the models. Over South America, data assimilation decreases tropospheric NO$_2$ columns by up to 25% in the biomass burning season in all the models, while the negative model

biases in the biomass-burning off-season are mostly removed.

Over Africa, the annual mean negative model biases are reduced from -0.32 to -0.11 × 10$^{15}$ molecules cm$^{-2}$ to -0.08 to 0.03 × 10$^{15}$ molecules cm$^{-2}$ (by 75–100%) over northern Africa, from -0.20 to -0.02 × 10$^{15}$ molecules cm$^{-2}$ to -0.03 to 0.02×10$^{15}$ molecules cm$^{-2}$  (by 77–100%) over central Africa, and from -1.05 to -0.89 × 10$^{15}$ molecules cm$^{-2}$ to -0.47 to 0.45 × 10$^{15}$ molecules cm$^{-2}$  (by 48–63%) over southern Africa. The bias reductions over central and southern Africa are large in austral

winter-spring. Some of the model negative biases (14–50 %, with a standard deviation of 12 %) remain over southern Africa in austral winter. The inadequate corrections of tropospheric NO$_2$ columns could be attributed to the insufficient model resolution, short chemical lifetime of NOx, and biases in the simulated chemical equilibrium state. Spatial resolutions higher than the MIROC-Chem-H resolution (1.1° x 1.1°) would be useful to represent emissions and pollutants over individual sources.

**4.3 CO**

Fig. 10 compares the latitudinal variations of surface CO concentration against the WDCGG observations from 59 stations. All the models underestimate the zonal and annual mean CO concentrations by 25–70 ppb in the NH extratropics and by 10–60 ppb in the tropics (expect for MIROC-Chem-Chem-H), as in most of other CTMs (Shindell et al., 2006). In the SH extratropics, GEOS-Chem underestimates surface CO by 8–15 ppb, whereas MIROC-Chem and MIROC-Chem-H

overestimate it by 5–20 ppb. Data assimilation reduced most of the model biases for the globe, except for the remaining negative model biases in the tropics in GEOS-Chem and AGCM-CHASER. The different analysis results of CO at high latitudes could mainly reflect differences in atmospheric transport from mid latitudes among the model. The effect of data assimilation is limited because of the lack of measurements at high latitudes.

Fig. 11 compares the seasonal variation of surface CO for the selected stations.  All the models well captured the observed

seasonal variations, except for relatively low temporal correlations (Table 5) over Barbados (r=0.58–0.85) and Ascension (r=0.72–0.94). In the NH extratropics, for most stations, all the models reveal too low CO throughout the year, with larger biases in boreal winter than in summer in the models. The summertime negative biases are largest in GEOS-Chem. In the tropics, the rapid increases in CO associated with biomass burning, e.g., in October over Barbados and in September over



Ascension, are underestimated by all the models. In the SH extratropics, a large multi-model spread in the simulated CO exists in austral winter-spring, likely due to the different representation of poleward transport.

The reductions in the model negative bias in the NH owing to data assimilation can be found throughout the year, with annual mean bias reductions of 65–76% for Barrow, 41–74% for Cold Bay, and 57–94% for Iceland, with MIROC-Chem-H
exhibiting smaller reductions. The insufficient corrections in MIROC-Chem-H suggest the need to optimize the settings for the assimilation of total column retrievals for the higher resolution system. Further efforts are clearly needed for improving the CO analyses in MIROC-Chem-H. The negative model biases are also reduced at NH low latitudes, i.e., by 68–94% for Bermuda, 48–97% for Midway, and 22–63% for Mauna Loa. Over Barbados, data assimilation corrects the timing of the maximum (in March) and minimum (in August) concentrations and improved the temporal correlation from 0.58–0.85 to
0.75–0.85 in all the models, whereas the observed peak in October is not represented by all the systems. In the SH, the multi-model spread is greatly reduced by data assimilation, while showing improved agreements with the observations except for excessively high concentrations over Ascension in June-July in MIROC-Chem-H.

The vertical gradients of CO differ largely among the models (Fig. 12a), with the largest decrease in the annual tropical mean concentrations with height in GEOS-Chem from the lower to upper troposphere. The sharp decrease could be
associated with weaker deep convection. In addition, the larger OH concentrations in GEOS-Chem (Fig. 12b) suggest stronger chemical destruction in the middle and upper troposphere. In models other than GEOS-Chem, the tropical mean concentrations of CO show a clear maximum around 200 hPa. After data assimilation, the CO gradient became even larger in GEOS-Chem, in association with a large increase of OH in the upper troposphere. In other models, the data assimilation introduced sharper decreases in CO from about 850 to 600 hPa, as a consequence of the enhanced chemical destructions (i.e.,
the increased OH) at those levels. The increase in OH by data assimilation is largest in MIROC-Chem-H than in other models in the middle and upper troposphere, which have influenced the vertical profile of CO substantially.

In summary, the tropical annual mean CO gradient between the surface and 400 hPa is decreased by 1–7%, whereas the annual mean OH concentration is increased by 7–20% in the lower troposphere and 15–120% in the middle and upper troposphere in all the models. Therefore, the multi-constituent data assimilation provides strong constraints on the vertical
profiles of CO and other species mainly through substantial changes in OH. Changes in OH are further discussed in Section 4.4. It is also suggested that, even after the multi-constituent data assimilation, the representations of the vertical profiles can differ among the models, reflecting both the different model configurations, e.g., in terms of deep convection and chemical reaction rates, and the lack of direct observational constraints on the vertical profiles.

## 4.4 OH

Because of the simultaneous assimilation of multiple-species data, the global distribution of various species, including OH, is modified considerably in the assimilation systems. The concentration of OH is directly related to the concentrations of species determining the primary production (ozone and $H_2O$), removal (CO and $CH_4$), and regeneration of OH (NOx). Fig. 13 compares the global distributions of annual and tropospheric mean OH concentrations. The multi-model comparisons





reveal common characteristics such as higher concentrations in the tropics than in the extratropics and enhanced concentrations over central Africa, Indian Ocean, south and Southeast Asia, and tropical Atlantic. As summarized in Table 6, the simulated OH is higher in GEOS-Chem and AGCM-CHASER than other models over most of the major polluted areas such as eastern China, India, western Europe, India, Southeast Asia, and Africa. The multi-model standard deviation of OH

is large over central Africa, northern India and around the Himalayas, Malay peninsula, western United States, Brazil, and over the southern tropics such as over the eastern Pacific and northern Australia (right-top figure in Fig. 13). The zonal mean OH from the tropics to subtropics is lower in MIROC-Chem-H than in other models by approximately 30–45% (Fig. 14). The zonal mean OH shows a strong latitudinal gradient around the subtropics. The ratio of OH in the NH tropics–subtropics (Equator–30°N) to the NH mid-latitudes (30°N–60°N) ranges from 1.42 (GEOS-Chem) to 1.71 (MIROC-Chem).

Data assimilation largely modified global OH distributions in all the systems. The analyzed OH fields and data assimilation increments are often regionally localized, which demonstrates the importance of accurately representing different chemical regimes and local emissions for each region, for estimation of both regional and global OH distributions. The annual mean OH is increased in the SH extratropics by 10–25% in GEOS-Chem and AGCM-CHASER and by 30–50% in MIROC-Chem and MIROC-Chem-H, probably because of the increased ozone. MIROC-Chem-H shows large increases in OH by 20–40%

over Africa, Southeast Asia, tropical Pacific, and central and South America, associated with the increased ozone and decreased CO. The NH exhibits large inter-model differences in OH increments, decreasing in GEOS-Chem by 10–40% with large increments over east Asia, the United States, and Europe and increasing in MIROC-Chem and MIROC-Chem-H by 15–30 % and by 20–40 % over the continents, respectively. The negative increments in GEOS-Chem are likely associated with the increased CO and decreased NOx, whereas the positive increments in MIROC-Chem and MIROC-Chem-H could

be attributed to the increased ozone and increased NOx. The NH tropics–subtropics (Equator–30°N) to mid-latitudes (30°N–60°N) ratio of OH is increased in all the models by 1–15%, with a largest increase in GEOS-Chem (from 1.42 to 1.64).

Because of the data assimilation, the multi-model spread of OH is reduced by 24–58% over the major polluted areas of the globe such as over Europe (44%), China (31%), the United States (41%), and central Africa (50%) and South America (58%). At local scale, the multi-model spread is reduced largely over central eastern Africa (up to 55%), associated with

adjustments made to biomass burning plumes, and over Indonesia (up to 40%) and the western US (up to 55%), corresponding to large changes in local NOx and CO emissions and consequently in ozone productions. The improved multi-model consistency suggests that the multi-constituent data assimilation provides a more similar representation of the tropospheric chemistry system, by removing model errors in the relevant species in the individual systems.

As summarized in Table 7, the north-to-south gradient of the tropospheric OH (averaged below 300 hPa) decreased owing

to data assimilation in all the models, i.e., from 1.29–1.36 (1.32 ± 0.03) to 1.17–1.23 (1.19 ± 0.03), as similarly suggested by our previous analysis (Miyazaki et al., 2015). The simulated NH/SH ratio of OH simulated from the four models is in the range of 1.28 ± 0.10 in the ACCMIP multi-model estimates (Naik et al., 2013), whereas the values from the data assimilation runs are significantly lower. The data assimilation estimates are in better agreements with an observational estimate (0.97±0.12) obtained using methyl chloroform observations (Patra et al., 2014). The significant changes in the global OH





distributions, which are common to all the models, are important in propagating the observational information between various species and modulating the chemical lifetimes of many species, thus improving emission inversion. The simultaneous optimization of emissions and concentrations was essential to modify the global OH distributions. The increases (by 1–32 %) in the global mean OH concentrations by data assimilation in all the models, with the multi-model

mean values of $1.12\pm0.18 \times 10^6$ mol/cm$^3$ in the control runs and $1.26\pm0.10 \times 10^6$ mol/cm$^3$ in the data assimilation as summarized in Table 7, suggest overestimated CH$_4$ lifetimes in the model simulations.

## 5 Estimated emissions

### 5.1 NOx emissions

As summarized in Table 8 and shown in Fig. 15, the global total NOx emissions are increased by 12% in GEOS-Chem,

40% in AGCM-CHASER, 25% in MIROC-Chem, and 30% in MIROC-Chem-H due to data assimilation. The a posteriori global total emissions vary from 39.1 TgN (GEOS-Chem) to 51.9 TgN (AGCM-CHASER) with the multi-model mean of 47.6±5.8 TgN. The regional NOx emissions are increased in the models other than GEOS-Chem over the United States (with annual regional total emission increases of 10–22%), eastern China (2–34%) and over western Europe (7–23%). The a posteriori emissions over eastern China in these models (5.8–6.4 TgN) are closer to the HTAP-v2 2010 inventory (5.7 TgN)

than those from EDGAR v4.2 (4.2 TgN). The emissions over Europe are largely increased in MIROC-Chem-H (by 23%) and AGCM-CHASER (by 24%). In GEOS-Chem, the emissions are decreased over most parts of eastern China (by 21% for regional total emissions), the United States (by 9%), and western Europe (by 21%), where the a posteriori emissions are obviously lower than the other estimates. As shown in Fig. 16, the multi-model mean of the a posteriori emissions shows strong NOx emissions over major polluted areas, while the multi-model spread is large for eastern China, the eastern United

States, around Mexico city, western Europe, and South Africa. The multi-model spread of the regional NOx emissions is smaller than the assumed a priori emission uncertainty (i.e., by 40 %) for all the polluted areas (Table 8).

For biomass burning areas, the emissions are increased in the all the models by 17–25% over southeast Asia, by 13–30% over northern Africa, and by 4–39% over central Africa. The positive increments over northern and central Africa are smallest in MIROC-Chem-H, likely due to the use of updated biomass burning emission inventories (GFED v4) as well as

updated NO$_2$ retrievals. The a posteriori emissions for the biomass burning areas are similar between the four systems: 0.6–0.8 TgN (16% standard deviation) for southeast Asia, 2.9–3.2 TgN (4%) for northern Africa, and 2.2–2.8 TgN (10%) for central Africa. Over South Africa, the emissions are increased by 29–50% in all the systems, with a large multi-model spread of the a posteriori emissions (0.4–0.9 TgN, 31% standard deviation).

The seasonal variations in NOx emissions are largely modified by data assimilation for many regions, with common

features for all of the four systems (Fig. 17). Over eastern China, the emissions in early summer (June) and winter (November–January) are enhanced in all the systems, which could be associated with emissions from soils and the use of wintertime heating, respectively. The magnitude of the summertime enhancement differs among the models, which could





reflect the different chemical lifetime of NOx under strong photolysis conditions. Over the United States and Europe, large enhancements in late spring and early summer and subsequently in the seasonal amplitude are commonly found in all the systems. Also, the timing of maximum emissions in summer moves forward by a few months (from one to two months over eastern China and Europe and from two to three months over the United States) due to data assimilation in all the systems,

likely due to underestimated soil emissions in early summer, which has also been suggested by Oikawa et al. (2015).

Over India, the a posteriori emissions reveal strong increases from April to June in all the systems, which is likely associated with open biomass burning that is not represented by the bottom-up inventories (Venkataraman et al., 2006). Over Southeast Asia, the emissions are mostly increased throughout the year in all the systems, with large increases in the biomass burning season (boreal spring), except in MIROC-Chem-H. Over South America, the emissions in the biomass-burning season

(August–September) are decreased by 30–50 % due to data assimilation in all the systems. The negative increments suggest an overestimation of emissions by forest fires in dry conditions in the GFED v2, v3, and v4 inventories, as similarly suggested by Castellanos et al. (2014) for the GFED v3 inventory. In contrast, the emissions are increased in the biomass-burning off-season by 30–60% in all the systems.

Over northern Africa, in the biomass burning season (boreal winter), fire-related emission factors in the GFED v3 inventory

(AGCM-CHASER, MIROC-Chem) are suggested to be too low by 20–30%, whereas those in the GFED v2 (GEOS-Chem) and v4 (MIROC-Chem-H) inventories are too high by 50% and by 10%, respectively. The multi-model consistency is high throughout the year over northern Africa. Over central Africa, the emissions in the biomass burning season (July–September) are increased by 30–45 % from the GFED v2 and v3 inventories and decreased by 20 % from the GFED v4 inventory.

The differences in the a posteriori emissions could be explained by the different model configurations, such as the chemical

lifetime of NOx, vertical mixing, lightning NOx sources, and model resolutions for many regions. The obtained inter-model differences are generally larger for industrialized areas (12–31%) than biomass burning areas (4–21%), suggesting the substantial influences of different urban chemistry configurations and/or model settings for anthropogenic NOx emissions (e.g., $NO_2$:NO ratio). Large uncertainties in chemical NOx loss have the strong effects on the simulated NOx lifetime and the accuracy of top-down NOx source inversion (Lin et al., 2012; Stavrakou et al., 2013).

As discussed in Section 3.4, the $NO_2$ response to NOx emissions ($\frac{\Delta NO_2}{\Delta ENOx}$) is stronger in GEOS-Chem than in other models probably associated with a weaker chemical NOx loss. This suggests that the same levels of tropospheric $NO_2$ columns can be explained by smaller amounts of NOx emissions, and this could explain the lower a posterior NOx emissions evaluated in this model with respect to other models. The multi-model differences in simulated NOx levels could also explain parts of the diversity in model ozone response to NOx emissions (Section 3.4). In addition, processes such as vertical mixing and

lightning NOx productions are strongly model-dependent and influence the responses of $NO_2$ to NOx emissions. Meanwhile, the updated $NO_2$ retrievals were assimilated only in MIROC-Chem-H, whereas the diurnal emission variability was optimized from data assimilation in MIROC-Chem and MIROC-Chem-H. These differences could also lead to model





dependence on emission estimates and model responses to the updated emissions. To fully understand the inter-model differences of the a posteriori emissions, their influence needs to be explored.

## 5.2 CO emissions

The global total CO emissions are increased by 50% in GEOS-Chem, 51% in AGCM-CHASER, 12% in MIROC-Chem, and 25% in MIROC-Chem-H due to data assimilation, with a large diversity in the estimated global total emissions (943.3–1376.9 TgCO, with 16% multi-model standard deviation), as summarized in Table 8. The CO emissions are increased by 18–119% over eastern China, 9–122% over the United States, and 37–146% over Europe in all the models, suggesting significant underestimations of anthropogenic CO emissions in the bottom-up inventories used as a priori emissions. Using the same a priori emission data sets, the positive increments are larger in AGCM-CHASER than in MIROC-Chem over eastern China, western Europe, and the United States, likely associated with underestimated (or overestimated) chemical production (destruction), as similarly discussed by Jiang et al. (2015). In fact, AGCM-CHASER reveals relatively high OH concentrations corresponding to large CO emissions (c.f., Table 6) over these regions. The multi-model spread of the a posteriori emissions is large over these industrialized regions (13–32%), with largest spreads over central eastern China (Fig. 16).

The a posteriori emissions exhibit a wintertime peak over eastern China in the models other than MIROC-Chem and over Europe other than GEOS-Chem (Fig. 18). Stein et al. (2014) commonly found that large corrections are needed for CO emissions in winter–spring seasons for industrialized areas. Because the chemical destructions are weak in these seasons, the results suggest underestimations in the bottom-up inventories rather than model errors in OH. Meanwhile, the distinct differences in the seasonality as well as mean strength of the a posteriori emissions highlight strong model dependence of CO emission estimations for the anthropogenic emission regions.

Over India, a pronounced peak in boreal spring is commonly introduced, and the a posteriori emissions show similar seasonality between NOx and CO in all the systems. Over southeast Asia, the annual total emissions are decreased by 3–11% in all the models, with an enhanced multi-model discrepancy in the biomass burning season. Over South America, the annual total emissions are decreased by 27–41% in the models except for MIROC-Chem-H, with large reductions in the biomass burning season. The results suggest a common overestimation problem in fire-related emission factors for both CO and NOx (c.f., Section 5.1) in GFED v2 and v3 over South America. In African regions, although the analyzed seasonal variations are similar, the annual total emissions reveal large discrepancies among the models (20–35%). In comparison with the averaged values in other models, the estimated emissions are larger by 28% in GEOC-Chem over northern Africa, by 27% in MIROC-Chem-H over central Africa, and by 40% in GEOS-Chem over southern Africa. The multi-model spread of the a posteriori emissions is large over major biomass burning regions, such as eastern Central Africa, northern Thailand, and Amazon (Fig. 16). The substantial inter-model differences highlight the importance of chemistry and dynamics in understanding the carbon budget over these regions.



The inter-model differences in data assimilation adjustments and a posteriori emissions are generally larger for CO than for NOx, which can be associated with different representations of atmospheric transports such as convective transport and vertical mixing (e.g., Jiang et al., 2015) because of the longer chemical lifetime of CO. Also, differences in the chemical production of CO from the oxidation of NMHCs and the chemical lifetime of CO could lead to large multi-model

discrepancies in CO simulations and emission estimates, as similarly discussed by Gaubert et al. (2016). Thus, the differences in various factors can enhance the multi-model discrepancies in the a posteriori CO emissions.

Our results suggest requirements for further development of the CO emission optimization framework to obtain more consistent estimates, for instance, by using a longer assimilation window and a larger ensemble size. The data assimilation windows employed (2–6 hours) are clearly insufficient to optimize surface CO emissions using remote measurements with

considering the influence of atmospheric transports. The estimated CO emissions were also sensitive to the choice of other parameters such as localization length and covariance inflation factor, while optimal values of these parameters are expected to differ among the models manly associated with different representations of atmospheric transport among the models. Optimizing these parameters for individual models would thus also be important. Meanwhile, adding observational constraints, for instance, on NMHCs emissions from formaldehyde measurements (e.g., Stavrakou et al., 2009), and

considering inter-species correlations (e.g., between NOx and CO) would help to improve the data assimilation analysis and multi-model consistency. Some of the increments seem to be inadequate in MIROC-Chem-H, which could suggest different optimal settings requirements for the assimilation of total column retrievals and for higher resolution models.

## 6 Conclusions and discussion

We developed the MOMO-Chem framework to integrate a portfolio of forward CTMs (GEOS-Chem, AGCM-CHASER,

MIROC-Chem, MIROC-Chem-H) in a state-of-the-art ensemble Kalman filter data assimilation system. The data assimilation was used to simultaneously optimize both chemical concentrations and emissions of multiple species through ingestion of a suite of measurements (ozone, $NO_2$, CO, $HNO_3$) from multiple satellite sensors. The framework was used to demonstrate the importance of the performance of forecast models for tropospheric chemistry data assimilation and to provide multi-model integrated information on the tropospheric chemistry system.

The forecast performance of the models differed for many species because of the different model configurations. In the absence of data assimilation, the multi-model discrepancies and forecast model errors for ozone against the ozonesonde observations were obvious, with annual mean biases ranging from -5.1 to 1.4 ppbv (from -6.2 to -0.7 ppbv) in the lower troposphere and from -4.0 to 16.1 ppbv (from 2.8 to 20.5 ppbv) in the middle and upper troposphere at NH (SH) mid latitudes. Tropospheric $NO_2$ columns are largely underestimated by the models other than GEOS-Chem over major polluted

areas, whereas the simulated column peaks in over biomass burning areas are largely biased. For CO, all the models underestimated surface concentrations in the NH by 20–80 ppb.



Multi-constituent assimilation greatly improved the multi-model consistency and the level of agreements with independent measurements. In comparison with the ozonesonde measurements, the annual mean bias is reduced by about 40–80% in the NH, by 50–90% in the tropics, and 45–95% in the SH in the middle and upper troposphere, while reducing the multi-model spread of annual mean ozone by 20–60% in the NH and 30–85% in the SH. Data assimilation also reduced the model biases in tropospheric $NO_2$ columns by more than 40% for both major industrialized and biomass burning areas while improving the seasonal variations. The model negative biases of CO in the NH are also reduced by about 40–95% in all the models. These results demonstrate that harnessing the current observing system provides sufficient constraints to greatly reduce the influences of model errors and to provide the consistent concentration analysis.

The multi-model comparisons of tropospheric OH reveal common features of global distributions but with obvious differences in mean concentration levels among the models. Data assimilation increments for OH differ largely among the models, decreasing in GEOS-Chem by 10–40% over east Asia, the United States, and Europe and increasing in MIROC-Chem and MIROC-Chem-H over most parts of the NH by 15–40%. In spite of the different increments, the multi-constituent data assimilation reduced the multi-model spread by about 25–60% over major polluted areas, while the north-to-south hemispheric ratio is reduced in all the models from 1.32±0.03 to 1.19±0.03. These results suggest that the multi-constituent data assimilation framework can be used to provide a common representation of the tropospheric chemistry system that is less dependent on individual model performance.

The MOMO-Chem framework provides possible uncertainty ranges in the a posteriori emissions in the current data assimilation framework due to model errors, which are quantified in 4–31% for NOx and 13–35% for CO regional emissions. Meanwhile, the multi-model analysis commonly suggests potential problems in the bottom-up emission inventories, such as underestimation of soil NOx emissions in early summer at NH mid-latitudes, underestimations of open biomass burning emissions in spring over India, and overestimation of emissions by forest fires in dry conditions over South America. For NOx emissions, the large inter-model discrepancies are attributable to the chemical lifetime of NOx, vertical mixing, lightning NOx sources, and model resolution. For CO emissions, the a posteriori emission differences are largely attributable to different representations of atmospheric transport, such as convective transport and vertical mixing, as well as chemical destruction and production and the use of a short assimilation window. The larger discrepancy for CO emissions than for NOx emissions suggest the need to further develop the CO emission optimization framework, for instance, by using a longer assimilation window and a larger ensemble size.

The response of surface $NO_2$ and ozone concentrations to NOx emission perturbations is largely different among the models. A stronger ozone response could help to reduce model errors more efficiently through changes in the model ozone equilibrium state from the emission optimization. The multi-constituent framework allows us to evaluate model ozone responses in a realistic condition while considering possible error ranges in precursor's emissions. The ozone and emission analysis increment information obtained using the optimized emissions can be used as a diagnostic to quantify model sensitivities related to chemistry and transport. Thus, a systematic investigation of model ozone response and analysis increment in the multi-constituent data assimilation framework could benefit evaluation of future prediction of chemistry-



climate system as a hierarchical emergent constraint (Bowman et al., 2018). By using the multi-model integrated fields from MOMO-Chem, we estimated the surface concentration responses to NOx emissions in the NH to be largest in January for $NO_2$ (2.0 ppb/($10^{-11}$ kgNm$^{-2}$s$^{-1}$)) and in August for ozone (2.4 ppb/($10^{-11}$ kgNm$^{-2}$s$^{-1}$)). The estimated ozone response was larger in the tropics than in the NH, implying that any latitudinal shifts in NOx emissions from the extratropics to the tropics would lead to increases in global tropospheric ozone. The obtained results also suggest that the multi-model integrated fields could provide fundamentally different chemical relationships than those in the individual models, which would inform chemical predictions through relationships such as emergent constraints. This example demonstrates the unique capability of MOMO-Chem for various applications.

In summary, the MOMO-Chem framework can be used to generate an ensemble of data assimilation analyses, and to provide integrated unique information on the tropospheric chemistry system including precursor's emissions, while directly accounting for structural uncertainty. Meanwhile, the framework provides uncertainty ranges in data assimilation analyses including the a posteriori emissions due to model errors. The information on the uncertainty obtained from the multi-model framework could be used to suggest requirements for the development of the individual models and observations. To obtain highly consistent data assimilation fields, increasing observational constraints and/or optimization of model parameters, such as VOCs emissions, would be needed. Also, improving background error information (e.g., by using multi-model ensembles), considering inter-species emission correlations, and increasing the ensemble size would be useful to improve the performance of the individual data assimilation systems. Comparing different data assimilation methods, such as EnKF vs 4D-VAR, would also be important to investigate whether we are able to produce a consistent data assimilation analysis that is independent of both the data assimilation scheme and forecast model performance.

**Author contributions**

KM and KWB designed the study. KM, TW, and KY developed the data assimilation code and set up the data assimilation experiments. KS developed the model code. KM performed the model simulations and data assimilation experiments. KM and KWB prepared the manuscript with contributions from all co-authors.

**Acknowledgement**

We acknowledge the use of data products from the NASA AURA, EOS Terra, and Aqua satellite missions. We also acknowledge the free use of tropospheric $NO_2$ column data from the SCIAMACHY, GOME-2, and OMI sensors from http://www.qa4ecv.eu and www.temis.nl. This work was supported through JSPS KAKENHI grant numbers 15K05296, 26220101, 26287117, 16H02946, 18H01285, the Environment Research and Technology Development Fund (2-1803) of the Ministry of the Environment, Japan, and by the Post-K computer project Priority Issue 4 - Advancement of meteorological and global environmental predictions utilizing observational Big Data. The Earth Simulator was used for simulations as





"Strategic Project with Special Support" of Japan Agency Marine-Earth Science and Technology. Part of the research was carried out at the Jet Propulsion Laboratory, California Institute of Technology, under a contract with the National Aeronautics and Space Administration.

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



**Table 1: Summary of the forecast models used in this study**

| | GEOS-Chem | AGCM-CHASER (TRC-1) | MIROC-Chem | MIROC-Chem-H (TCR-2) |
|---|---|---|---|---|
| Horizontal resolution | 2°x2.5° | 2.8°x2.8° | 2.8°x2.8° | 1.1°x1.1° |
| Vertical resolution | 47 layers to 0.1 hPa (hybrid) | 32 layers to 4 hPa (sigma) | 32 layers to 4 hPa (hybrid) | 32 layers to 4 hPa (hybrid) |
| Forecast model | GEOS-Chem v9 (adjoint v35) | CCSR/NIES/FRCGC AGCM-CHASER | MIROC-Chem | MIROC-Chem |
| Chemistry | 43 species, 318 reactions | 47 species, 88 reactions | 92 species, 262 reactions | 92 species, 262 reactions |
| Meteorological data | GEOS-5 | Nudged to NCEP-DOE/AMIP-2 | Nudged to ERA-Interim | Nudged to ERA-Interim |
| Assimilated data | OMI $NO_2$ (DOMINO2), SCIAMACHY $NO_2$ (DOMINO2), TES ozone (v5) MOPITT CO (v6 NIR) MLS ozone & $HNO_3$ (v3.3) | OMI $NO_2$ (DOMINO2), SCIAMACHY $NO_2$ (DOMINO2), TES ozone (v5) MOPITT CO (v6 NIR) MLS ozone & $HNO_3$ (v3.3) | OMI $NO_2$ (DOMINO2), SCIAMACHY $NO_2$ (DOMINO2), TES ozone (v5) MOPITT CO (v6 NIR) MLS ozone & $HNO_3$ (v3.3) | OMI $NO_2$ (QA4ECV), SCIAMACHY $NO_2$ (QA4ECV), TES ozone (v6) MOPITT CO (v7J) MLS ozone & $HNO_3$ (v3.3) |
| A priori emissions | EDGAR, NEI2008, RETRO, GFED2 | EDGAR 4.2 GFED 3.1, GEIA | EDGAR 4.2, GFED 3.1, GEIA | HTAP v2, GFED 4, GEIA |
| State vector | Concentrations of 43 species + emissions (NOx, CO, LNOx) | Concentrations of 35 species + emissions (NOx, diurnal variability, CO, LNOx) | Concentrations of 35 species + emissions (NOx, diurnal variability, CO, LNOx) | Concentrations of 35 species + emissions (NOx ,diurnal variability, CO, $SO_2$, LNOx) |
| Reference (Forecast model) | Henze et al. (2007) | Sudo et al. (2002) | Watanabe et al. (2011) | Sekiya et al. (2018) |
| Reference (Data assimilation) | This study | Miyazaki et al. (2012a, 2012b, 2014, 2015) | Miyazaki et al. (2017) | Miyazaki et al. (2019) |





**Table 2 Linear regression of changes in surface NOx emissions (in $10^{-11}$ kgNm$^{-2}$s$^{-1}$) and surface concentrations of ozone and NO$_2$ (in ppb) by data assimilation in May 2007 over areas with NOx emission changes greater than $5\times10^{-13}$ kgNm$^{-2}$s$^{-1}$ in the four models. The results for regions without strong NOx emission changes (greater than $3\times10^{-11}$ kgNm$^{-2}$s$^{-1}$, that could suffer from dilution effects) are shown in brackets.**

|  |  | Slope (ppb/($10^{-11}$ kgNm$^{-2}$s$^{-1}$)) | Intercept (ppb) | Correlation |
|---|---|---|---|---|
| Ozone | GEOS-Chem | 1.2 (1.8) | 2.2 (2.2) | 0.34 (0.34) |
|  | AGCM-CHASER | 1.5 (3.6) | 3.6 (3.0) | 0.42 (0.45) |
|  | MIROC-Chem | 1.0 (2.5) | 3.0 (2.7) | 0.35 (0.42) |
|  | MIROC-Chem-H | 0.6 (1.3) | 4.1 (3.9) | 0.25 (0.42) |
| NO$_2$ | GEOS-Chem | 0.80 | 0.01 | 0.93 |
|  | AGCM-CHASER | 0.56 | 0.01 | 0.96 |
|  | MIROC-Chem | 0.54 | 0.01 | 0.94 |
|  | MIROC-Chem-H | 0.44 | 0.04 | 0.94 |



**Table 3: Annual mean bias of the mean ozone concentrations (in ppbv) between the data assimilation or control run (in brackets) and the ozonesonde observations from the WOUDC network for 850–500 hPa and 500–200 hPa in ppb for five latitudinal bands, SH high latitudes (55°S–90°S), SH mid-latitudes (15°S–55°S), tropics (15°S–15°N), NH mid-latitudes (15°N–55°N), and NH high latitudes (55°N–90°N). The results are shown for individual models and multi-model mean (mean±1 sigma).**

| | | 90°S–55°S | 55°S–15°S | 15°S–15°N | 15°N–55°N | 55°N–90°N |
|---|---|---|---|---|---|---|
| 850-500 hPa | GEOS-Chem | (-2.3) 7.4 | (-0.7) 2.2 | (-1.2) 0.0 | (1.4) 3.4 | (-2.6) 2.3 |
| | AGCM-CHASER | (-2.2) 2.4 | (-2.0) 2.7 | (-1.0) 3.6 | (-2.0) 5.2 | (-4.7) 4.3 |
| | MIROC-Chem | (-4.6) 0.9 | (-1.5) 1.9 | (2.2) 4.6 | (-1.3) 2.6 | (-4.7) 1.7 |
| | MIROC-Chem-H | (-4.5) 3.6 | (-6.2) 1.0 | (-5.8) 2.3 | (-5.1) 1.9 | (-3.6) 3.2 |
| | Multi-model | (-3.4±2.0) 4.0±2.5 | (-2.5±2.5) 1.6±1.1 | (-1.4±3.1) 1.6±1.8 | (1.8±2.7) 3.3±1.6 | (-3.9±1.4) 2.2±1.7 |
| 500-200 hPa | GEOS-Chem | (8.7) 4.7 | (10.5) 3.0 | (1.3) -0.3 | (-4.0) -0.9 | (-29.9) -9.2 |
| | AGCM-CHASER | (29.9) 1.6 | (8.6) -1.9 | (-3.2) 0.3 | (0.3) -1.9 | (-13.5) -6.2 |
| | MIROC-Chem | (28.7) 1.0 | (20.5) -1.3 | (0.4) -0.2 | (16.1) -0.2 | (12.9) -6.1 |
| | MIROC-Chem-H | (24.8) 1.4 | (2.8) 0.3 | (-11.8) 3.0 | (4.1) -2.5 | (29.7) -5.5 |
| | Multi-model | (23.0±10.7) 4.0±2.8 | (10.6±7.2) 1.1±2.3 | (-3.3±5.3) 1.7±1.5 | (4.1±8.6) -1.5±2.1 | (-0.2±24.0) -7.8±2.9 |





**Table 4: Annual mean bias and temporal correlation of regional mean tropospheric NO$_2$ columns: the data assimilation minus the satellite retrievals from OMI in 10$^{15}$ molecules cm$^{-2}$. The results of the control run are also shown in brackets. The results are shown for eastern China (110–123°E, 30–40°N), Europe (10°W–30°E, 35–60°N), USA (70–125°W, 28–50°N), South America (50–70°W, 20°S–Equator), northern Africa (20°W–40°E, Equator–20°N), central Africa (10–40°E, Equator–20°S), and southern Africa (25–34°E, 22–31°S). The results are shown for individual models and multi-model mean (mean±1 sigma).**

|  |  | E China | E USA | Europe | India | SE Asia | S America | N Africa | C Africa | S Africa |
|---|---|---|---|---|---|---|---|---|---|---|
| Bias | GEOS-Chem | (0.13) | (0.29) | (0.60) | (-0.20) | (-0.15) | (-0.02) | (-0.11) | (-0.11) | (-0.89) |
|  |  | -0.16 | 0.11 | 0.21 | -0.05 | 0.05 | 0.02 | 0.00 | 0.00 | -0.46 |
|  | AGCM-CHASER | (-2.07) | (-0.51) | (-0.82) | (-0.27) | (-0.44) | (-0.06) | (-0.32) | (-0.39) | (-1.05) |
|  |  | -0.69 | -0.18 | -0.41 | -0.12 | -0.18 | -0.02 | -0.08 | -0.09 | -0.45 |
|  | MIROC-Chem | (-1.59) | (-0.26) | (-0.32) | (-0.15) | (-0.30) | (-0.11) | (-0.30) | (-0.39) | (-0.93) |
|  |  | -0.60 | -0.14 | -0.35 | -0.09 | -0.18 | -0.03 | -0.07 | -0.09 | -0.34 |
|  | MIROC-Chem-H | (-0.37) | (-0.31) | (-0.56) | (-0.14) | (-0.56) | (-0.19) | (-0.23) | (-0.23) | (-0.96) |
|  |  | -0.39 | -0.13 | -0.24 | -0.02 | -0.13 | 0.01 | 0.03 | 0.01 | -0.47 |
|  | Multi-Model | (-1.04±0.94) | (-0.34±0.11) | (-0.58±0.20) | (-0.18±0.06) | (-0.36±0.18) | (-0.10±0.07) | (-0.24±0.09) | (-0.28±-0.14) | (-0.96±0.07) |
|  |  | -0.46±0.23 | -0.09±0.13 | -0.20±0.28 | -0.07±0.04 | -0.11±0.11 | -0.01±0.02 | -0.03±0.05 | -0.04±0.05 | -0.43±0.06 |
| T-Corr | GEOS-Chem | (0.97) | (0.70) | (0.95) | (0.87) | (0.96) | (0.97) | (0.90) | (0.96) | (0.38) |
|  |  | 1.00 | 0.86 | 0.97 | 0.94 | 0.99 | 1.00 | 0.98 | 0.99 | 0.92 |
|  | AGCM-CHASER | (0.95) | (0.83) | (0.95) | (0.66) | (0.95) | (0.94) | (0.94) | (0.96) | (0.96) |
|  |  | 1.00 | 0.97 | 0.97 | 0.98 | 1.00 | 1.00 | 0.99 | 1.00 | 1.00 |
|  | MIROC-Chem | (0.96) | (0.80) | (0.94) | (0.20) | (0.92) | (0.91) | (0.96) | (0.96) | (0.98) |
|  |  | 0.99 | 0.97 | 0.97 | 0.99 | 1.00 | 0.99 | 0.99 | 1.00 | 1.00 |
|  | MIROC-Chem-H | (0.98) | (0.58) | (0.90) | (0.86) | (0.86) | (0.98) | (0.99) | (0.99) | (0.98) |
|  |  | 1.00 | 0.94 | 0.98 | 0.98 | 0.97 | 0.99 | 0.99 | 1.00 | 0.98 |
|  | Multi-Model | (0.97±0.01) | (0.73±0.11) | (0.94±0.02) | (0.64±0.31) | (0.92±0.05) | (0.95±0.03) | (0.95±0.04) | (0.97±0.02) | (0.83±0.30) |
|  |  | 1.00±0.01 | 0.94±0.05 | 0.97±0.01 | 0.97±0.02 | 0.99±0.01 | 1.00±0.01 | 0.99±0.01 | 1.00±0.01 | 0.98±0.04 |



**Table 5: Annual mean bias and temporal correlation of surface CO. Units are ppb. The observations used are the WDCGG observations. The results of the control run are also shown in brackets. The results are shown for individual models and multi-model mean (mean±1 sigma).**

| | | Barrow | Cold Bay | Iceland | Bermuda | Midway | Mauna Loa | Barbados | Ascension | Cape Grim | Showa | South Pole |
|---|---|---|---|---|---|---|---|---|---|---|---|---|
| Bias | GEOS-Chem | (-47) -11 | (-48) -12 | (-40) -3 | (-34) 7 | (-44) -22 | (-33) -13 | (-30) -13 | (-12) -7 | (-11) -6 | (-10) -6 | (-6) -3 |
| | AGCM-CHASER | (-43) -14 | (-50) -21 | (-45) -12 | (-32) -2 | (-35) -1 | (-28) -16 | (-19) -16 | (-19) -18 | (-2) -1 | (-5) -5 | (-1) 3 |
| | MIROC-Chem | (-23) -8 | (-41) -24 | (-30) -13 | (-19) -6 | (-19) -1 | (-23) -18 | (-17) -17 | (-4) -12 | (6) -2 | (5) -4 | (10) 14 |
| | MIROC-Chem-H | (-35) -34 | (-30) -28 | (-23) -23 | (-13) -13 | (-18) -20 | (-6) -9 | (-2) -9 | (10) 4 | (18) 7 | (17) 3 | (23) 9 |
| | Multi-Model | (-37±11) -17±12 | (-42±9) -21±7 | (-35±10) -13±8 | (-25±13) -3±12 | (-29±12) -11±9 | (-23±12) -14±4 | (-17±12) -14±4 | (-11±6) -8±9 | (-9±7) -1±5 | (-9±6) -3±4 | (-10±9) 6±7 |
| T-Corr | GEOS-Chem | (0.94) 0.90 | (0.99) 0.85 | (0.99) 0.95 | (0.95) 0.95 | (0.99) 0.94 | (0.99) 0.96 | (0.85) 0.75 | (0.72) 0.90 | (0.96) 0.96 | (0.97) 0.96 | (-0.24) 0.20 |
| | AGCM-CHASER | (0.80) 0.97 | (0.54) 0.84 | (0.74) 0.94 | (0.89) 0.97 | (0.70) 0.96 | (0.85) 0.92 | (0.58) 0.85 | (0.94) 0.96 | (0.75) 0.75 | (0.99) 0.99 | (0.44) -0.13 |
| | MIROC-Chem | (0.74) 0.80 | (0.71) 0.77 | (0.89) 0.95 | (0.97) 0.96 | (0.88) 0.94 | (0.88) 0.96 | (0.64) 0.83 | (0.89) 0.90 | (0.96) 0.96 | (0.97) 0.97 | (0.35) 0.27 |
| | MIROC-Chem-H | (0.88) 0.90 | (0.95) 0.90 | (0.98) 0.96 | (0.95) 0.92 | (0.96) 0.94 | (0.96) 0.96 | (0.80) 0.81 | (0.78) 0.71 | (0.98) 0.95 | (0.98) 0.99 | (0.47) 0.69 |
| | Multi-Model | (0.84±0.09) 0.89±0.07 | (0.80±0.21) 0.84±0.05 | (0.90±0.12) 0.95±0.01 | (0.94±0.03) 0.95±0.02 | (0.88±0.13) 0.95±0.01 | (0.92±0.07) 0.95±0.02 | (0.72±0.13) 0.81±0.04 | (0.83±0.10) 0.86±0.11 | (0.91±0.10) 0.91±0.10 | (0.98±0.01) 0.98±0.01 | (0.26±0.33) 0.26±0.34 |





**Table 6: Annual and regional mean OH concentration at 700 hPa, Units are $10^6$ mol/cm$^3$. The results of the control run are also shown in brackets. The results are shown for individual models and multi-model mean (mean±1 sigma). Changes in the multi-model spread due to data assimilation (Δspread, in %) are also shown.**

|  | E China | E USA | Europe | India | SE Asia | S America | N Africa | C Africa | S Africa |
|---|---|---|---|---|---|---|---|---|---|
| GEOS-Chem | 1.9 | 1.5 | 1.2 | 2.5 | 2.3 | 1.2 | 2.1 | 2.1 | 2.2 |
|  | (2.2) | (1.8) | (1.5) | (2.5) | (2.2) | (1.1) | (2.1) | (2.1) | (2.1) |
| AGCM-CHASER | 2.4 | 1.7 | 1.4 | 2.6 | 2.2 | 1.2 | 2.3 | 1.9 | 2.4 |
|  | (2.5) | (1.7) | (1.4) | (2.6) | (2.0) | (1.2) | (2.1) | (1.6) | (2.1) |
| MIROC-Chem | 2.1 | 1.5 | 1.3 | 2.4 | 2.1 | 1.3 | 2.4 | 1.9 | 2.1 |
|  | (1.9) | (1.3) | (1.1) | (2.1) | (1.8) | (1.1) | (2.0) | (1.5) | (1.7) |
| MIROC-Chem-H | 1.6 | 1.3 | 1.1 | 1.9 | 1.6 | 1.1 | 1.8 | 1.7 | 1.7 |
|  | (1.4) | (1.1) | (1.0) | (1.7) | (1.3) | (0.7) | (1.4) | (1.3) | (1.1) |
| Multi-model | 2.0±0.3 | 1.5±0.2 | 1.3±0.1 | 2.3±0.3 | 2.0±0.3 | 1.2±0.1 | 2.1±0.2 | 1.9±0.2 | 2.1±0.3 |
|  | (2.0±0.4) | (1.5±0.3) | (1.3±0.2) | (2.2±0.3) | (1.8±0.3) | (1.0±0.2) | (1.9±0.3) | (1.6±0.3) | (1.8±0.4) |
| Δspread [%] | -31 | -41 | -44 | -25 | -27 | -58 | -16 | -50 | -24 |



**Table 7: Interhemispheric gradient (NH/SH) and global mean (with area weight) concentration (in $10^6$ mol/cm$^3$) of tropospheric mean OH (averaged between the surface and 300 hPa) from the control and data assimilation runs. The results are shown for individual models and multi-model mean (mean±1 sigma).**

|  |  | GEOS-Chem | AGCM-CHASER | MIROC-Chem | MIROC-Chem-H | Multi-model |
|---|---|---|---|---|---|---|
| NH/SH ratio | Model | 1.30 | 1.36 | 1.29 | 1.31 | 1.29±0.03 |
|  | Assimilation | 1.17 | 1.23 | 1.18 | 1.21 | 1.18±0.03 |
| Global mean | Model | 1.31 | 1.23 | 1.13 | 0.82 | 1.12±0.18 |
|  | Assimilation | 1.31 | 1.31 | 1.34 | 1.09 | 1.26±0.10 |





**Table 8: Annual regional total NOx emissions of NOx (in TgNyr$^{-1}$) and CO (in TgCOyr$^{-1}$) obtained from the a priori emissions and a posteriori emissions for 2007. The multi-model mean and standard deviation ("spread") of the a posteriori emissions among the four systems (in %) is also shown.**

| | | E China | USA | W Europe | India | SE Asia | S America | N Africa | C Africa | S Africa | Globe |
|---|---|---|---|---|---|---|---|---|---|---|---|
| NOx | GEOS-Chem | 3.8 | 4.7 | 3.7 | 2.3 | 0.6 | 1.1 | 3.1 | 2.2 | 0.4 | 39.1 |
| | | (4.6) | (5.1) | (4.5) | (2.1) | (0.5) | (1.1) | (2.7) | (1.6) | (0.2) | (37.2) |
| | AGCM-CHASER | 6.4 | 7.2 | 5.1 | 3.0 | 0.8 | 0.9 | 2.9 | 2.4 | 0.9 | 51.9 |
| | | (4.2) | (5.6) | (4.1) | (2.6) | (0.6) | (1.3) | (2.1) | (1.6) | (0.5) | (39.2) |
| | MIROC-Chem | 5.8 | 6.3 | 4.4 | 3.0 | 0.8 | 1.2 | 3.0 | 2.6 | 0.8 | 49.1 |
| | | (4.2) | (5.6) | (4.1) | (2.6) | (0.6) | (1.3) | (2.1) | (1.6) | (0.5) | (39.2) |
| | MIROC-Chem-H | 5.8 | 6.1 | 4.8 | 2.9 | 0.6 | 1.5 | 3.2 | 2.8 | 0.7 | 50.4 |
| | | (5.7) | (5.5) | (3.7) | (3.1) | (0.5) | (1.5) | (2.8) | (2.7) | (0.5) | (42.4) |
| | Multi model ±spread [%] | 5.5±21% | 6.1±17% | 4.5±13% | 2.8±12% | 0.7±16% | 1.2±21% | 3.0±4% | 2.5±10% | 0.7±31% | 47.6±12% |
| CO | GEOS-Chem | 146.0 | 149.4 | 81.1 | 84.1 | 23.1 | 63.8 | 131.9 | 117.9 | 14.0 | 1376.9 |
| | | (123.6) | (67.3) | (35.0) | (67.9) | (23.8) | (87.2) | (123.4) | (82.4) | (4.5) | (916.1) |
| | AGCM-CHASER | 152.9 | 123.9 | 74.5 | 67.3 | 26.8 | 42.9 | 110.2 | 95.2 | 11.9 | 1275.9 |
| | | (69.7) | (59.0) | (30.3) | (63.9) | (29.1) | (73.3) | (88.4) | (87.1) | (5.8) | (843.5) |
| | MIROC-Chem | 128.6 | 97.0 | 54.9 | 53.8 | 24.5 | 51.1 | 66.7 | 96.7 | 6.5 | 943.3 |
| | | (69.7) | (59.0) | (30.3) | (63.9) | (29.1) | (73.3) | (88.4) | (87.1) | (5.8) | (843.5) |
| | MIROC-Chem-H | 177.5 | 71.2 | 37.4 | 76.8 | 16.7 | 68.2 | 102.1 | 142.8 | 7.7 | 1108.8 |
| | | (130.0) | (65.2) | (27.3) | (67.4) | (18.8) | (64.1) | (98.6) | (93.2) | (6.7) | (888.0) |
| | Multi model ±spread [%] | 151.3±13% | 110.4±31% | 62.0±32% | 70.5±19% | 22.8±19% | 56.5±21% | 102.7±26% | 113.2±20% | 10.0±35% | 1176.2±16% |



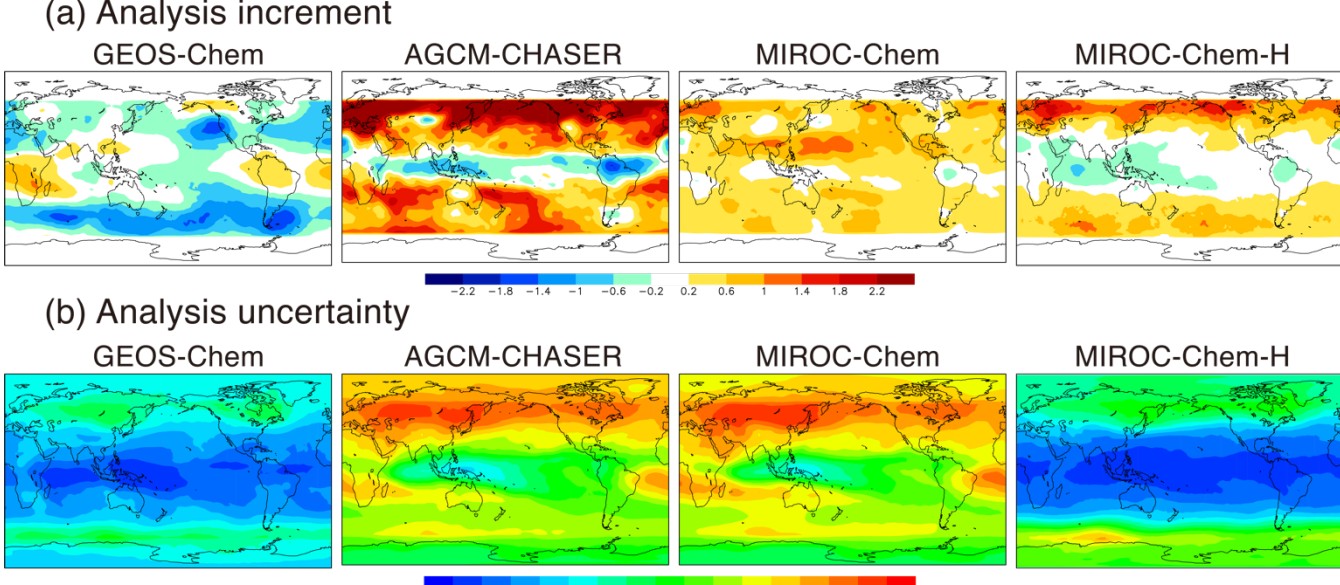

**Figure 1: Spatial distributions of (a) analysis increment (in ppbv/day) and (b) analysis uncertainty (in ppb) of ozone at 500 hPa averaged over 2007 from the four systems.**



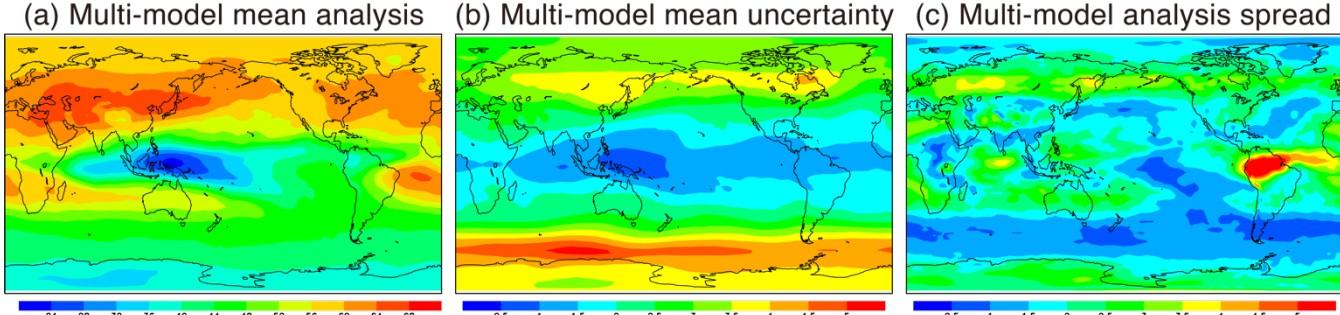

**Figure 2: Spatial distributions of the multi-model mean values of (a) data assimilation analysis and (b) its uncertainty of annual mean ozone at 500 hPa estimated using Eqs. (10) and (11), respectively. (c) shows the standard deviation (i.e., multi-model spread) of the annual mean ozone analysis among the four models.**





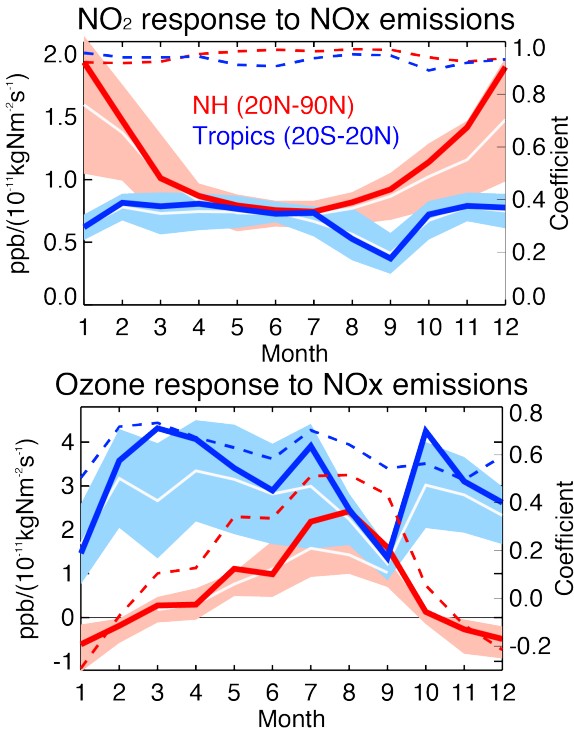

**Fig. 3: Time series of model response of surface ozone and NO₂ concentrations to NOx emissions estimated from linear regressions using the multi-model integrated fields in 2007 over areas with NOx emission changes greater than 5×10⁻¹³ kgNm⁻²s⁻¹ for the Northern hemisphere (20°N–60°N, red line) and the tropics (20°S–20°N, blue line). The ±1σ deviation among the four models (i.e., model spread) is shown in light red for the NH and in light blue for the tropics. The multi-model mean value (i.e., an average of individual estimates) is shown by white lines.**





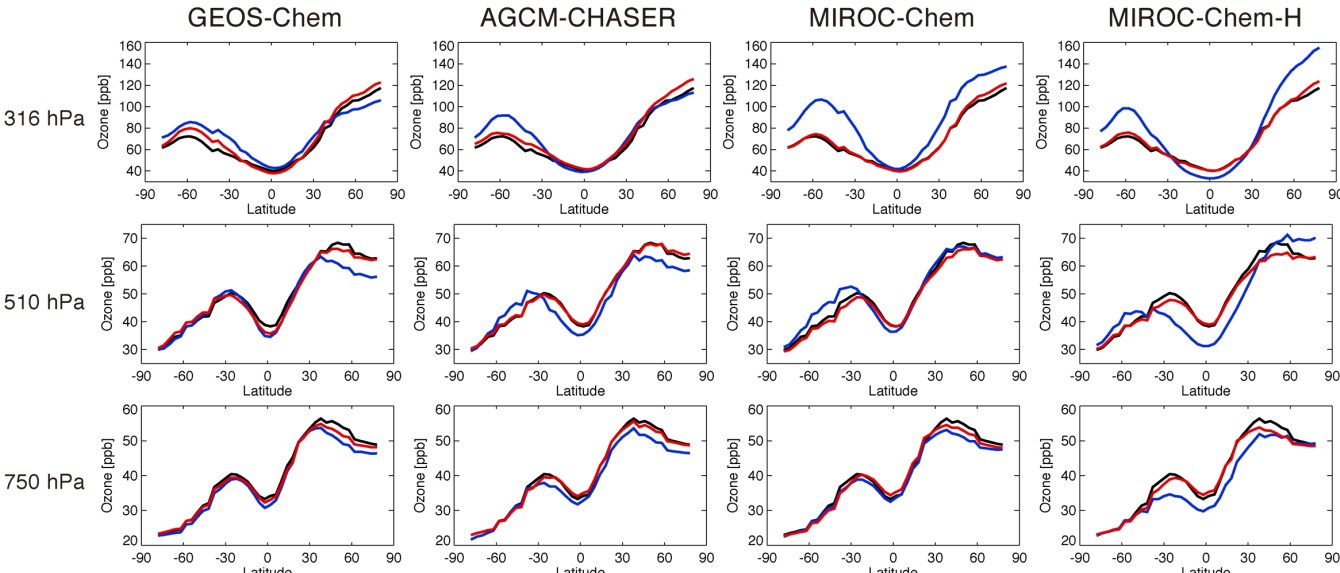

**Figure 4: Comparisons of latitudinal distributions of annual and zonal mean ozone concentrations between the TES measurements (black line), control runs (blue line), and data assimilation analyses (red line) at 316 hPa (upper panels), 510 hPa (middle panels), and 750 hPa (lower panels) in 2007 for the four systems.**





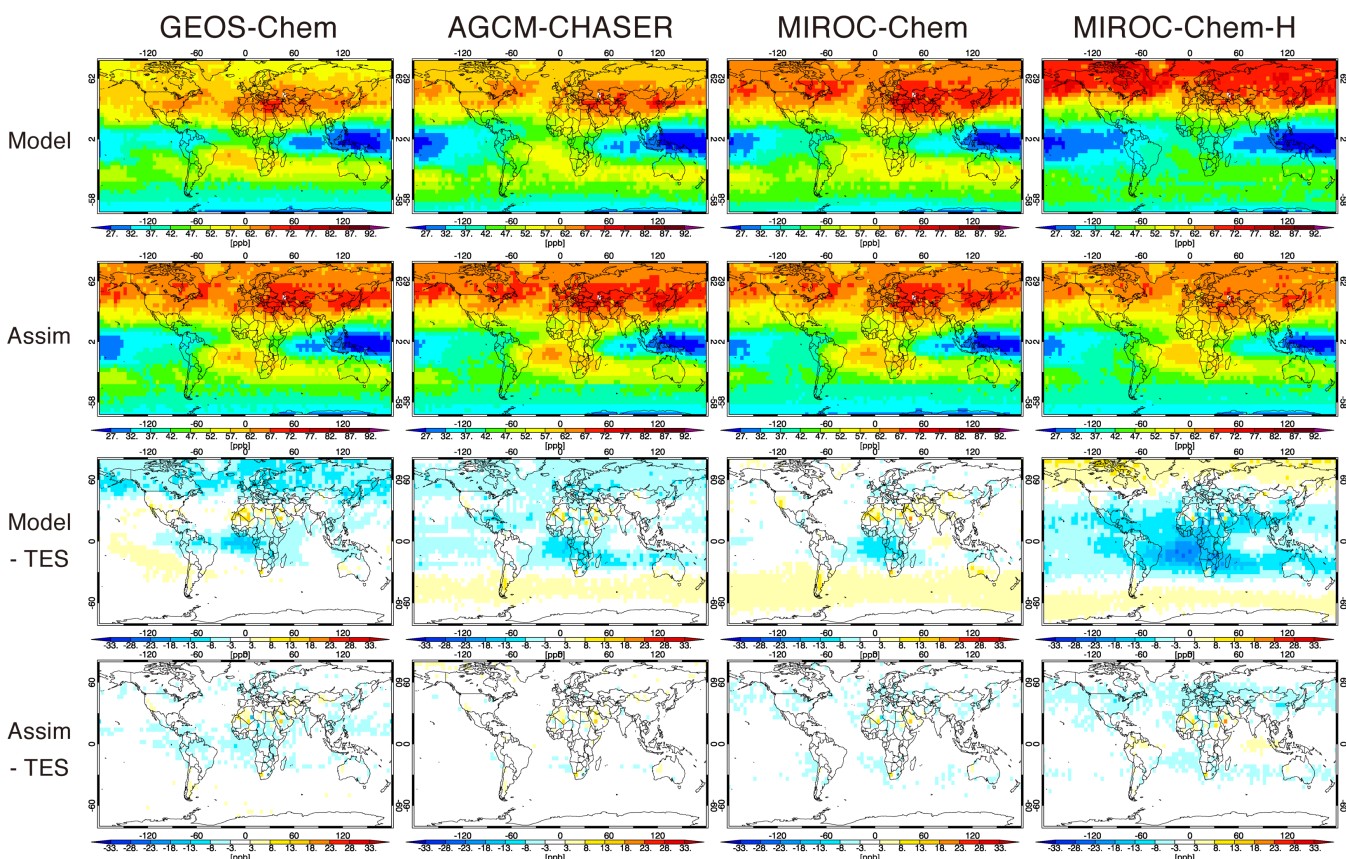

**Figure 5: Comparisons of the spatial distributions of annual mean ozone concentrations between the TES measurements, control runs, and data assimilation analyses at 510 hPa in 2007. Units are ppbv.**





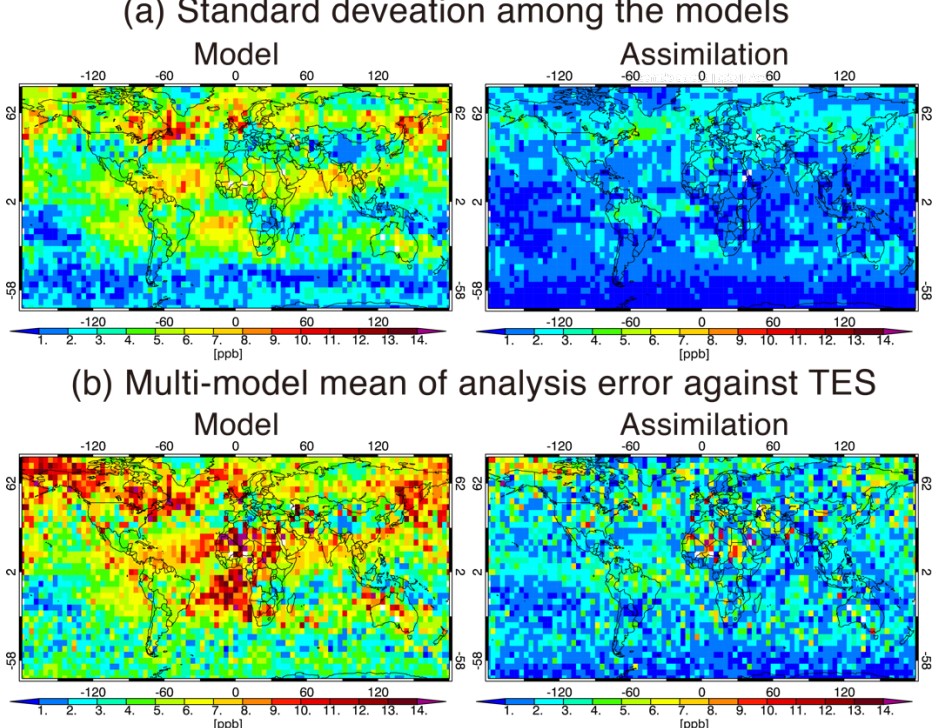

**Figure 6: (Top panels) Multi-model standard deviation of the data assimilation analysis with applying the TES AK at 510 hPa. (Bottom panels) Spatial distributions of multi-model mean (root-mean-square) values of error against TES measurements for the control runs (left) and data assimilation analyses (right) at 510 hPa.**



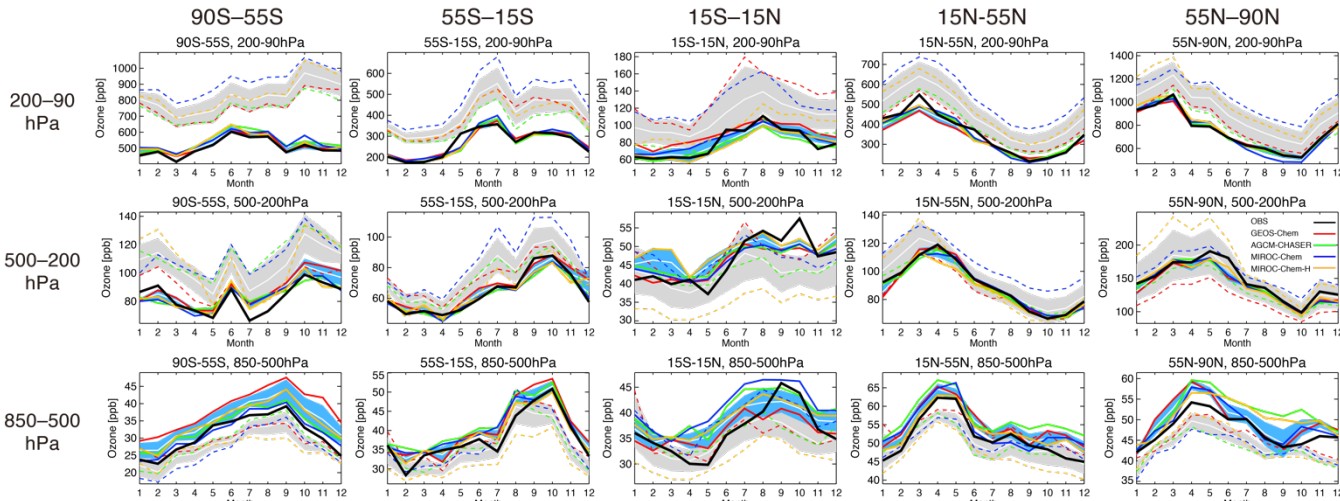

**Figure 7: Comparison of seasonal variation of ozone concentration between the ozonesonde observations (black solid line), model simulations (colored dotted lines), and data assimilation (colored solid lines) averaged between 90°S–55°S, 55°S–15°S, 15°S–15°N, 15°N–55°N, and 55°N–90°N for 2007. From top to bottom, results are shown for concentrations averaged over 80–200 hPa, 200–500 hPa, and 500–850 hPa. The ±1σ deviation among the four models (i.e., model spread) is shown in grey for the control runs and in light blue for the data assimilation results.**



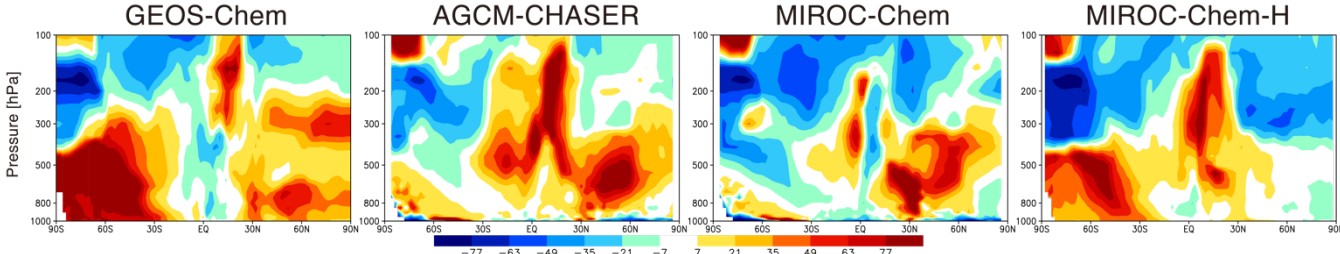

**Figure 8: Latitude-pressure cross section of changes in seasonal amplitude of zonal mean ozone concentrations (in ppbv) due to data assimilation (data assimilation minus control runs), as estimated from the maximum minus minimum values of the monthly mean ozone concentrations.**



**Figure 9: Time series of regional monthly mean tropospheric NO₂ columns (in 10¹⁵ molecules cm⁻²) from the satellite retrievals**
**(black for OMI QA4ECV and gray for OMI DOMINO v2), control runs (colored dotted lines), and data assimilation analysis**
**(colored solid lines) for 2007. The model simulation and data assimilation results are obtained at the local overpass time of the**
**retrievals by applying the averaging kernel of OMI DOMINO v2 for GEOS-Chem, CHASER, and MIROC, and of OMI QA4ECV**
**for MIROC-H, respectively, corresponding to the assimilated measurements. The multi-model standard deviations are not shown**
**because of the use of different assimilated measurements in the individual systems.**





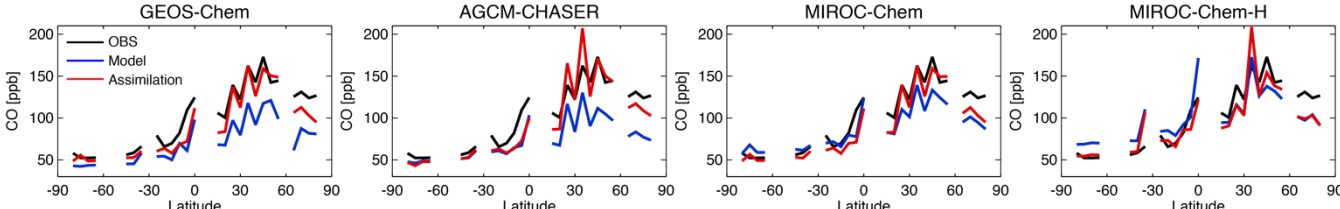

**Figure 10:** **Latitudinal distributions of zonal mean surface CO concentrations (in ppbv) averaged over the WDCGG surface measurement sites from the observations (black), control runs (blue), and data assimilation analyses (red).**





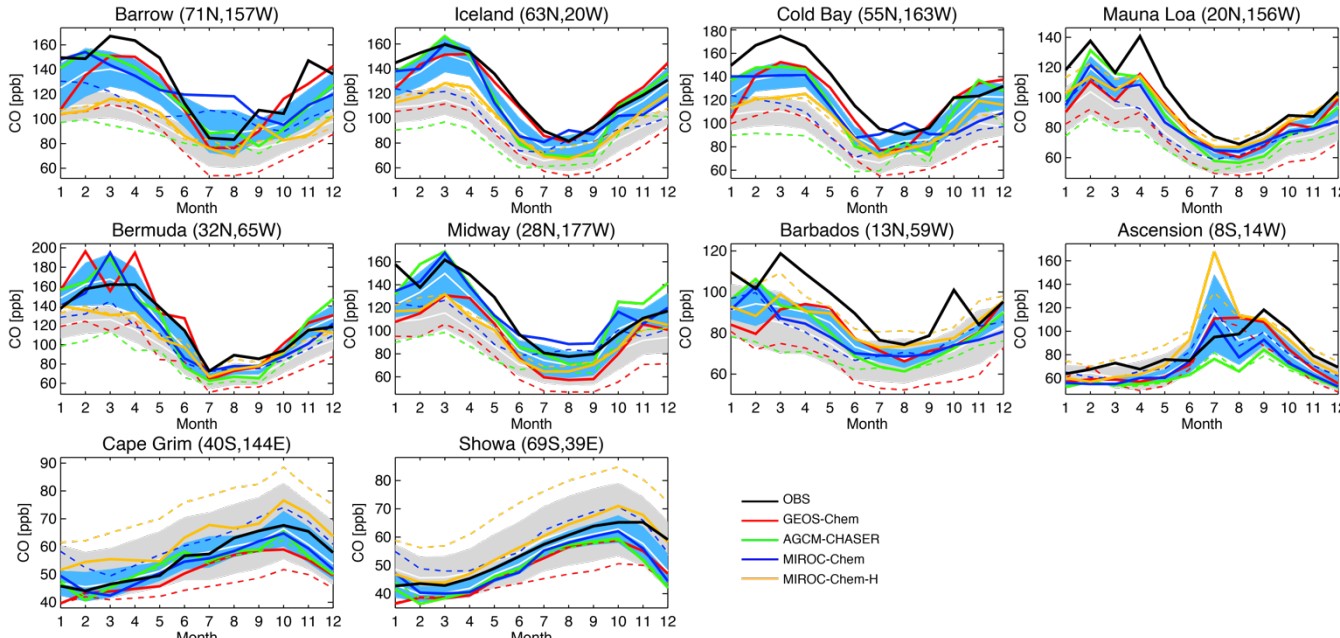

**Figure 11: Time series of monthly mean surface CO concentrations (in ppbv) from the WDCGG observations (black solid line), control runs (colored dotted lines), and data assimilation analyses (colored solid lines). The ±1σ deviation among the four models (i.e., model spread) is shown in grey for the control runs and in light blue for the data assimilation results.**





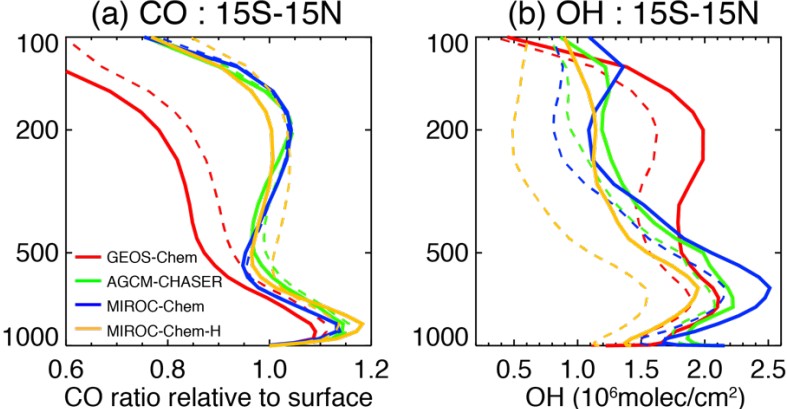

**Figure 12: Vertical profiles of (a) annual mean CO concentrations and (b) annual mean OH concentrations averaged between 15°S–15°S, obtained from the control runs (dotted lines) and data assimilation analysis (solid line). For CO (a), the relative ratio to the mean surface concentrations is shown. For OH (b), the unit is $10^6$ mol/cm$^3$.**



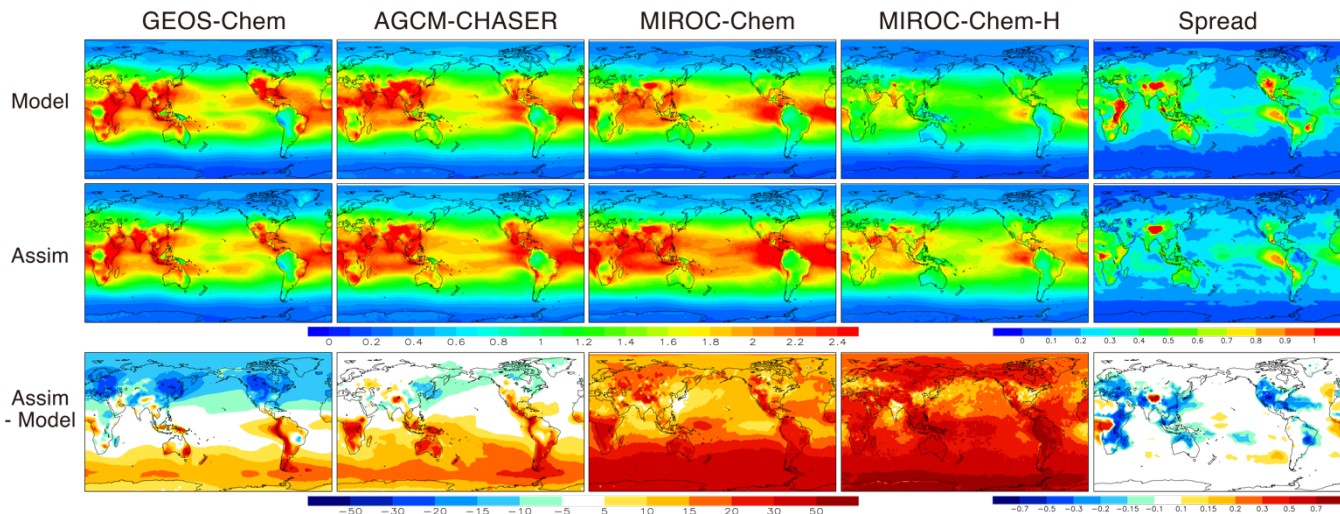

**Figure 13:** **Global distributions of annual mean OH concentrations (in $10^6$ mol/cm³) averaged over the troposphere from the control runs (upper panels), data assimilation analyses (middle panels), and differences between the data assimilation analyses and the control runs (bottom panels, in %). The figures on the right shows the standard deviation among the four systems for the control runs (right top) and data assimilation analyses (right middle). The right bottom figure shows the difference in the multi-model standard deviation between the data assimilation analyses and the control runs (in $10^6$ mol/cm³).**


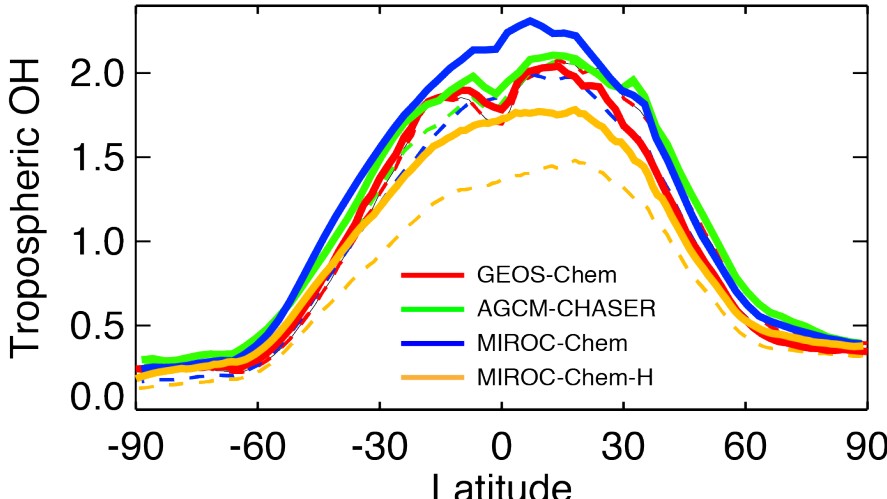

**Figure 14: Latitudinal distributions of annual and zonal mean OH concentrations (in $10^6$ mol/cm³) averaged over the troposphere obtained from the control runs (dotted lines) and data assimilation analyses (solid lines) for the four systems.**





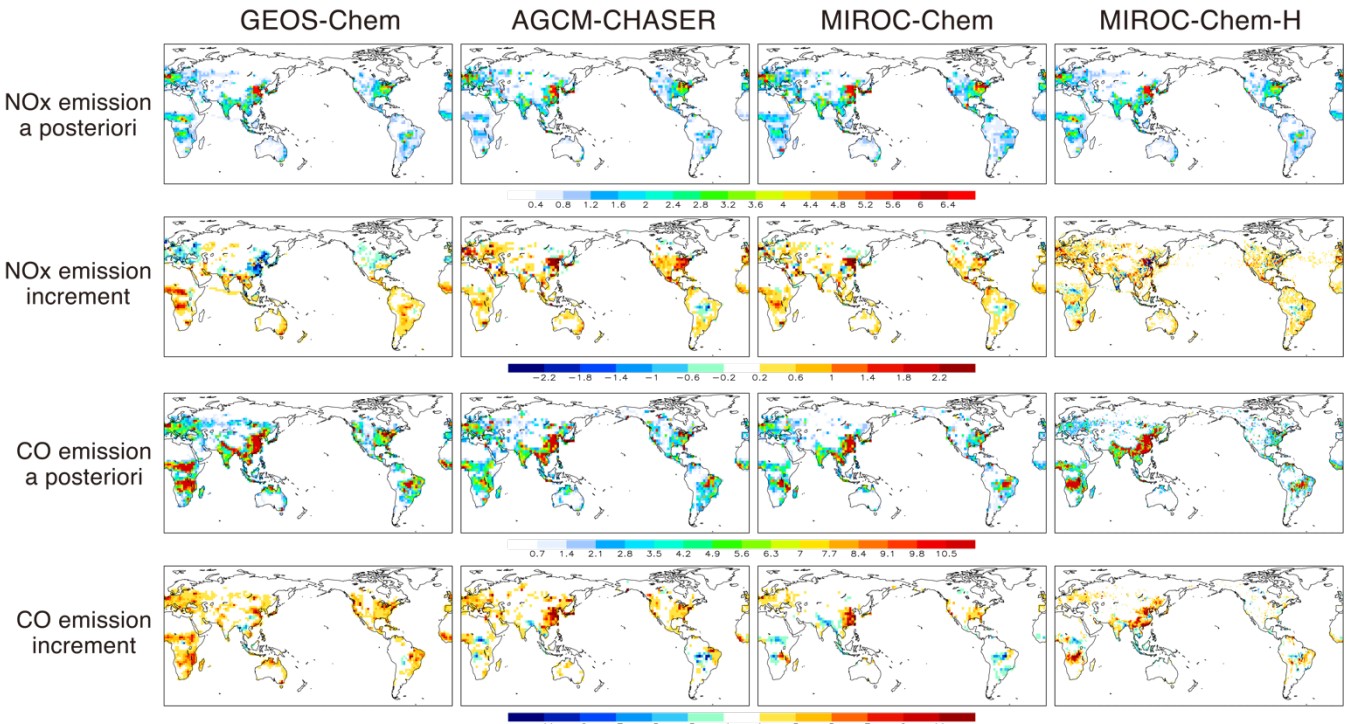

**Figure 15: Global distributions of annual mean surface NOx emissions (in 10⁻¹¹ kgN m⁻²s⁻¹) and surface CO emissions (in 10⁻¹⁰ kgCO m⁻²s⁻¹) for 2007. The a posteriori emissions and analysis increment (a posteriori minus a priori emissions) are shown.**





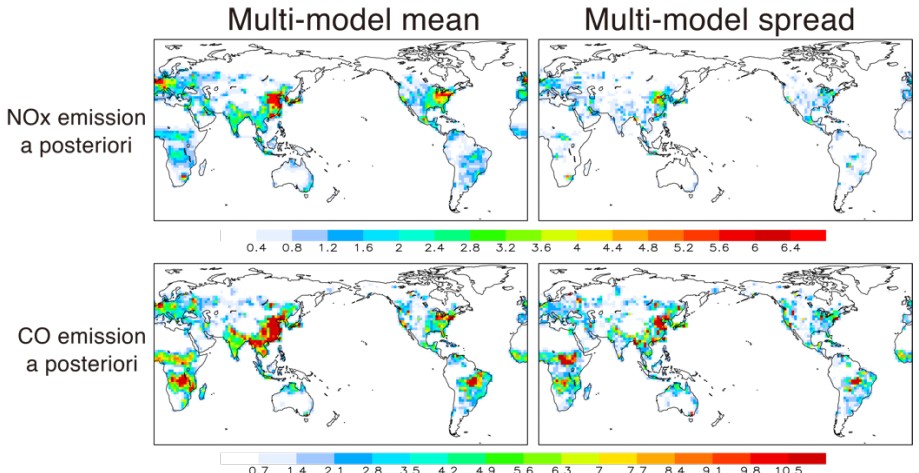

Figure 16: Same as Fig. 15, but for the multi-model mean and spread.





**Figure 17: Time series of monthly total regional surface NOx emissions (in TgNyr⁻¹) obtained from the a priori emissions (dotted lines) and the a posteriori emissions (solid lines) for 2007 for the four systems. AGCM-CHASER and MIROC-Chem use the same a priori emissions. The ±1σ deviation among the four models (i.e., model spread) is shown in grey for the control runs and in light blue for the data assimilation results.**





**Figure 18: Same as in Fig. 17, but for monthly total regional surface CO emissions (in TgCOyr⁻¹).**

