# Peer review of "Evaluation of a multi-model, multi-constituent assimilation framework for tropospheric chemical reanalysis"

_Atmospheric Chemistry and Physics, 2019_

## Referee Comment (RC1) · Anonymous Referee #1 · 25 Sep 2019

Major comments:

This study presents an unprecedented work consisting in applying the same data assimilation framework to different Chemical Transport Models. This allows to treat the different flavor of model error terms such as chemistry, transport and deposition schemes in a consistent manner. The study is well written and presents a comprehensive work. The evaluation against independent measurements is greatly appreciated.

The discussion on the model error is quite interesting, this could be further compared to the observations error, which has been effectively used in the analysis procedure. It is stated that the analysis error can be overestimated, which is not common in EnKFs

where the spread in the ensemble is usually too small. The ensemble spread and the analysis error of the model mean against TES can be as low as 1 ppb (Fig. 6). One can expect that the use of a multi-model estimate (inter-model differences in the spread) of the model error fits nicely the observation error. In general, throughout the paper, error bars can be added to the observations.

P9L25: "The state vector includes the chemical concentrations of various species as well as the surface sources of NOx and CO and LNOx sources". And P10L6: "We also applied covariance localization for different variables in the state vector (Kang et al., 2011), by setting the covariance among non- or weakly related variables to be zero." Could you be more specific and describe which observations are used to esti- mate which state variable? In particular in the case of the Ozone responses to NOx perturbation, it would be interesting to know which state variables are optimized while assimilating ozone.

Minor comments: For future work, you could look at the impact of assimilation on the O3-CO correlations and dO3/dCO enhancement ratios (Zhang et al. 2006).

P5L4: the family name of the author is "Olivier" not "Oliver" according to https://themasites.pbl.nl/tridion/en/themasites/edgar/publications/index-2.html

P9L24: The data assimilation settings were almost same among the systems as fol- lows. Should it be a "the" before same? this sentence can be improved.

P12L26 "lowover", is there a space missing ?

P15L12: "the large multi-model spreads (25–55%) suggest that individual models have large uncertainty in representing strong ozone productions, for instance, associated with VOCs emissions and chemistry." Is it really an uncertainty or is it just because the chemical regime is not in favor of a clear and linear relationship between Ozone and NOx emissions.

P15L16: "the mean of the individual model estimates (solid while lines)". I guess you

mean 'white line' instead of 'while line'.

P21 L34: "Fig.13 compares the global distributions of annual and tropospheric mean OH concentrations." How do you define tropospheric? – also in legend of Fig. 13.

Figure 3: You should define what the dashed lines are in legend.

Figure 5: you could replace 'model' by 'control'

Figure 6: "(a) Standard Deveation among the models"

References: Zhang, L., et al. (2006), Ozone-CO correlations determined by the TES satellite instrument in continental outflow regions, Geophys. Res. Lett., 33, L18804, doi:10.1029/2006GL026399.

---

## Referee Comment (RC2) · Anonymous Referee #2 · 11 Oct 2019

Overview:

The authors evaluate and synthesize the results of four global chemical tropospheric reanalyses for the year 2007. All reanalyses were produced with the same EnKF DA system, that includes the optimisation of emissions and assimilates satellite retrieval for multiple-species. The main difference between the reanalyses is that a different Chemistry Transport Model, each having its own resolution and emissions input, was used in the EnKF DA system for the production the reanalysis data set. The authors demonstrate that all systems perform well by showing that the four reanalyses of atmospheric concentrations are considerably closer to independent and assimilated observations

than corresponding control runs without assimilation. At the same time the authors find that posterior CO and NOx emissions (i.e. analysed) vary considerably between the four reanalyses because a different relation between CO and NOx emissions changes and CO, NO2 and O3 concentration changes is simulated by the different CTMs.

From the four analyses the authors calculate a "integrated" analysis as weighted average of the four analysis. The weights are the analysis uncertainties, expressed as standard deviation, derived from the EnKF DA system. This "integrated analysis" is used to calculate regression estimates of the response between concentration changes and emissions changes.

The evaluation of the re-analyses with observations is carried out for the 4 reanalysis and a "multi-model" analysis, which is an unweighted average of the four analysis but not the "integrated" analysis.

General comments:

The paper provides a consistent inter-comparison of state-of-the-art EnK chemical DA systems. Exploring the importance of the modelling approach, the authors show (i) that the analyses are an improved (compared to the models) representation of the atmospheric concentration for the assimilated variables and (ii) the limitation of current EnK approaches to come to consistent, i.e. model independent, posterior estimate of the emissions. The later point is of particular interest to the community and needs to be more emphasised.

Presenting the calculation of the weighted mean of the four analyses ("integrated reanalysis") as "Multi-mOdel Multi-cOnstituent Chemical data assimilation (MOMO-Chem) framework" sounds like an inflation of terms. But more importantly, it is misleading because the term framework implies to me that the four EnK DA system runs were connected in any way. After reading the paper carefully, I came to the conclusion that this seems not the case. (I hope it did not miss anything). I believe the averaging procedure could be carried for any set of KF DA analysis even without the need to

use the same EnKF software. I strongly recommend changing the title and modify the abstract accordingly.

Combining four analysis using their standard deviation estimate as weights, is of course an advantage of the analyses compared to model runs for which this uncertainty information is not available. On the other hand, the analysis error standard deviation is a derived quantity (from P and R) and not estimated using independent observations. Therefore, it would be beneficial to show that the weighted mean approach leads to better results than an unweighted mean ("multi model"), or perhaps even other approaches to represent ensembles such as the median or the choice of best performing system.

The validation section (4) only discusses the errors against independent observations of the 4 individual analyses and their unweighted mean ("multi-model") but not the weighted mean ("integrated). This seems not consistent with the intension of the paper and all the discussion before, which focuses on the merits of the weighted-mean ("integrated") analysis. A comparison of unweighted ("multi-model") vs. weighted ("integrated") would be an important aspect of the paper, which is missing. Tables 3 to 6 should also include the respective figures for the weighted ("integrated") means. Further, moving the validation section 4 (Validation results) before section 3 (Data assimilation statistics) seems advisable because having established trust in the data set by the validation could make the discussion on emission response more convincing.

The authors use the standard deviation to represent the variation between the four analyses. Given that 4 is a relatively small number and Gaussianity can not be assumed, using a different measure such as the range should be considered.

The paper is quite long. Presenting the four reanalysis for concentrations and emission response would be one very valid paper. Comparing the different approaches to combine the reanalyses could be a second paper.

Specific comments

Abstract:

p1 l14. Please consider not to introduce a complex acronym for an averaging procedure of independent data sets.

p1 l 21. The sentence starting " These improvements .." is difficult to understand. Improvement would imply a reduction in uncertainty of the emissions but this seems no the case. Please make a quantitative statement on the global or regional total of the prior and posterior emissions.

p1 l25. The last sentence is not a consequence of the statement before, but quite the opposite. It only shows how difficult the estimation of emissions is because of the complexities of atmospheric chemistry.

p1l 26. Please clarify "emerging constraints" in the abstract. Also why specifically the "integrated" reanalyses are better suited for this than the single re-analysis or the "multi-model" mean reanalysis.

p2 l20 These are climate runs over several years. In the data assimilation context, model errors are the errors in the forecast over the short assimilation window started from previous analysis. Is not clear that long-term errors are a predictor for the short-range forecast errors of the model. Inconsistencies between the analysis and the model equilibrium state and inadequate background error statistics (inflation factors etc.) may play a large role too, but do not effect the multi-year runs.

p3 l3 "a common data assimilation framework " , see my general comments. I think it is not a common data assimilation framework because the information from the error statistics (P) of each system is not used to come up with a better common analysis. Running different chemistry scheme as part of the ensemble would have been a common framework. The four systems are completely independent. You compare and average different analysis and emissions/concentration responses.

p3 l8 Please mention that the assimilated observations are not the same for all 4 reanalyses.

p4 l3 Please provide more details on the quantitative value of the inflation factors and any differences between the four applications.

P4 l19 "a common data assimilation framework to four CTM frameworks" better "the same data assimilating system using 4 different CTM".

p4 l20 Please include a general statement that the CTMs differ in emissions, chemistry scheme, resolution etc.

p4l 23 - p5 l24 The model descriptions vary to much in detail. In particular section 2.2.4 (MIROC-Chem) seems shorter than the others and does not have detailed information about the emissions. Please avoid too much duplication of information included in the text and provided in table 1. I recommend including the facts in the table and discuss specifics in the text.

p7 l3 S and A should be in a bold font as in the formula below

p7 l1o Please clarify this sentence. Biases in the observations, in particular if the instrument biases in NO2 and Ozone are not consistent, may severely degrade the performance. I would even speculate the a potential inconsistency between the NO2 and Ozone retrievals are a major reason for problems with the performance of chemical DA systems.

p8 l11 Please make a comment on the differences between the two MOPITT versions.

p9 l2 "We construct integrated data assimilation analysis using multiple models combined with multiple-species measurements." I am not sure if the word "integrated" implies what you do. You calculate a mean of 4 analyses, which were produced with the same DA system using each a different modelling approach. This is not integration or data assimilation as such but simply a weighted average. Please consider replacing the word ("integrated") throughout the text. Please make clear a distinction between the weighted and unweighted mean.

p9 l4 Please provide more details on the standard deviation field sigma_j. It would be helpful to include a formula in section 2.1 Also say if sigma_j is a field of the same time-space dimension as the analysis.

p9 l5 Please clarify what the spatial resolution of mean reanalysis is, given that the for re-analysis are carried out at different resolutions (table 5)

p9 l16 Please discuss here (or elsewhere) quantitatively the standard deviations to give an indication which model dominates the weighted-mean analysis.

p9 l13 Please clarify why this is not "meaningful"

p9 l26 The longer assimilation window for GEOS-Chem could have implications for the assimilation of NO2 and CO because their life time can be short and the diurnal cycle of the NOx emissions is very pronounced.

p11 l20 Please provide evidence for the fact that TES plays the largest role.

p12 l24 instead of "integration" use "mean"

p12 l31 The smaller standard deviation seems "trivial" given the formula of its calculation. Only the comparison with independent observation can demonstrate that the mean is a more reliable representation of the field.

p13 l16 Please provide more detail how the regression coefficients are calculated. Do they represent the mean over regions, month and all hours of the day? Please explain how the response is calculated for the weighted mean ("integrated") analysis.

p13 l29 Please clarify that sentence. The study comes up with a different response functions for each model but gives no guidance which is the correct one. Given the non-linearities a mean over conflicting responses might not be the most suited approach.

p14 l17 I would think "conflicting" is a better description then "unique" for the difference in the response

p15 l13 Please clarify" inter-model correlations"

p15 l29 Please clarify if the only difference between "mean" and "integrated" (formula 10) is the weighting with the analysis error standard deviations. If this is larger than the unweighted mean, this means a model with a larger "ozone response" had a smaller analysis error standard deviation. In this sense the standard deviation of the analysis error is also used as an indicator for the reliability of the "ozone response". Is this true an how can this be justified?

p11-p18 Please discuss also the errors of the weighted mean ("integrated") analysis and update the tables

p18 l10 Please clarify this sentence. What are the estimated mean errors (the biases?) and how do they relate to the analysis uncertainty, which is expressed as (bias corrected) standard deviation

p18 l13. Are you suggesting the AGCM-CHASER and MIROC-Chem should be given more weight in the weighted ("integrated") analysis.

p22 l22 The reduced multi-model spread for OH is an important finding and should presented more prominently. It shows that the assimilation of O3, NO2 and CO can improve the OH field because different models come to the similar results even without assimilation of OH observations.

p23 l11 Please give here also the respective numbers for the prior emissions.

p23 l20 The comparison seems not quite correct. The prior emissions are biased against each other, which will also strongly influence the posteriori spread.

p25 l5 Could these rather large and diverse correction of the CO emission indicate that the important impact of the VOC emissions on CO concentrations is not considered with the setup of EnKF DA system?

p28 l2 Please mention that these results were obtained by a regression approach.

p27l18 Please clarify uncertainty ranges in this context. Does it included that the biases or is it just the standard deviation. You should mention that the prior emissions had smaller differences between each other than a posteriori emissions.

p28 l5 "The obtained results also suggest that the multi-model integrated fields could provide fundamentally different chemical relationships than those in the individual models" I find this statement not convincing. A divergent response of the different model shows predominately that a robust conclusion can not be drawn from this multi-model approach. If only, it means we need more research to find more realistic chemical relationships.

Table 8 Please include multi-model mean and spread for prior emissions Caption: Please indicate what the number in brackets are

Figure 3: Indicate what the dashed lines represent

Figure 8: Correct spelling of "deveation"

Figure 15: The differences between the different a posteriori and prior estimates between the models could be shown better.

---

## Author Comment (AC1) · 5 Nov 2019

**Author's comments in reply to the anonymous referee for "Evaluation of a multi-model, multi-constituent assimilation framework for tropospheric chemical reanalysis" by K. Miyazaki et al.**

We want to thank the referees for the helpful comments. We have revised the manuscript according to the comments, and hope that the revised version is now suitable for publication. Below are the referee comments in italics with our replies in normal font.

**Reply to Referee #1**

*This study presents an unprecedented work consisting in applying the same data assimilation framework to different Chemical Transport Models. This allows to treat the different flavor of model error terms such as chemistry, transport and deposition schemes in a consistent manner. The study is well written and presents a comprehensive work. The evaluation against independent measurements is greatly appreciated. The discussion on the model error is quite interesting, this could be further compared to the observations error, which has been effectively used in the analysis procedure. It is stated that the analysis error can be overestimated, which is not common in EnKFs where the spread in the ensemble is usually too small. The ensemble spread and the analysis error of the model mean against TES can be as low as 1 ppb (Fig. 6). One can expect that the use of a multi-model estimate (inter-model differences in the spread) of the model error fits nicely the observation error. In general, throughout the paper, error bars can be added to the observations.*

Thank you for the helpful comments. We have added the following comments about the relationship between the observation error and the obtained results. Since the detailed observation error information are not provided for most of the independent observations, we have not added error bars in the figures.

In Section 2.4.1

"The observation error is 5-10 % between the surface and 30 km (Smit et al., 2007)."

In Section 4.1.2

"The obtained mean errors and multi-model spreads of the data assimilation analysis are comparable to or smaller than the ozonesonde observation errors (5-10 %). "

In Section 4.1.1

"The mean retrieval uncertainty of the TES measurements is typically between 5-10 ppb in the SH and 10-15 ppb in the NH, which are larger than the multi-model spread and the mean model errors after data assimilation."

In Section 4.2

"The OMI $NO_2$ super-observation error is typically about 20-50 % of the tropospheric $NO_2$ columns over polluted areas, which are comparable to or larger than the analysis error."

*P9L25: "The state vector includes the chemical concentrations of various species as well as the surface sources of NOx and CO and LNOx sources". And P10L6: "We also applied covariance localization for different variables in the state vector (Kang et al., 2011), by setting the covariance among non- or weakly related variables to be zero." Could you be more specific and describe which observations are used to estimate which state variable? In particular in the case of the Ozone responses to NOx perturbation, it would be interesting to know which state variables are optimized while assimilating ozone.*

The following sentences have been added:

"Concentrations of NOy species and ozone were optimized from TES ozone, OMI and SCIAMACHY $NO_2$, and MLS ozone and $HNO_3$ observations."

*Minor comments: For future work, you could look at the impact of assimilation on the O3-CO correlations and dO3/dCO enhancement ratios (Zhang et al. 2006).*

Thanks for the suggestions. I agree that ozone-CO correlations would be another interesting model response that could be measured using the MOMO-Chem framework. In the current data assimilation setting, we didn't optimize CO concentrations using ozone observations to avoid sampling errors (c.f., Section 2.6). To note the future possibility, the following sentences have been added:

"In addition, tropospheric ozone shows strong correlations with other species such as CO (Zhang et al., 2006) over regions such as continental outflow regions. The relationship can be included in the state vector to improve the tropospheric ozone analysis. The uncertainty information in the CO response to ozone obtained from the MOMO-Chem can be expected to provide useful information on model diagnostics and future predictions."

*P5L4: the family name of the author is "Olivier" not "Oliver" according https://themasites.pbl.nl/tridion/en/themasites/edgar/publications/index-2.html*

Corrected.

*P9L24: The data assimilation settings were almost same among the systems as follows. Should it be a "the" before same? this sentence can be improved.*

Rewritten as

"Almost the same data assimilation settings were used for the four systems as follows."

*P12L26 "lowover", is there a space missing ?*

Added.

*P15L12: "the large multi-model spreads (25–55%) suggest that individual models have large uncertainty in representing strong ozone productions, for instance, associated with VOCs emissions and chemistry." Is it really an uncertainty or is it just because the chemical regime is not in favor of a clear and linear relationship between Ozone and NOx emissions.*

It is possible that the chemical regime differs among the models because of, for instance, different level of VOCs. Also, chemical and transport mechanisms affecting ozone formation can differ among the models. To clarify it the sentence has been rewritten as follow:
"the large multi-model spreads (25–55%) suggest that individual models have large uncertainty in representing strong ozone productions, for instance, associated with VOCs emissions and chemistry that could results in different chemical regimes."

*P15L16: "the mean of the individual model estimates (solid while lines)". I guess you mean 'white line' instead of 'while line'.*

Corrected

*P21 L34: "Fig.13 compares the global distributions of annual and tropospheric mean OH concentrations." How do you define tropospheric? – also in legend of Fig. 13.*

The tropospheric mean OH concentrations were calculating by averaging OH from the surface to 300 hPa, which is explained in the revised manuscript. Note that the averages between the surface to 300 hPa and between the surface to the tropopause made only small differences.

*Figure 3: You should define what the dashed lines are in legend.*

Defined.

*Figure 5: you could replace 'model' by 'control'*

In all the figures, we labeled "Model" rather than "Control". Although either is fine, we would like to keep the consistent labeling (Model).

*Figure 6: "(a) Standard Deveation among the models"*

Collected.

---

## Author Comment (AC2) · 5 Nov 2019

**Author's comments in reply to the anonymous referee for "Evaluation of a multi-model, multi-constituent assimilation framework for tropospheric chemical reanalysis" by K. Miyazaki et al.**

We want to thank the referees for the helpful comments. We have revised the manuscript according to the comments, and hope that the revised version is now suitable for publication. Below are the referee comments in italics with our replies in normal font.

*Reply to Referee #2*

*General comments:*
*The paper provides a consistent inter-comparison of state-of-the-art EnK chemical DA systems. Exploring the importance of the modelling approach, the authors show (i) that the analyses are an improved (compared to the models) representation of the atmospheric concentration for the assimilated variables and (ii) the limitation of current EnK approaches to come to consistent, i.e. model independent, posterior estimate of the emissions. The later point is of particular interest to the community and needs to be more emphasised.*

Thank you for the helpful comments. We have endeavored to address your helpful concerns and comments in the revised manuscript. Please see additional replies to your comments below.

*Presenting the calculation of the weighted mean of the four analyses ("integrated reanalysis") as "Multi-mOdel Multi-cOnstituent Chemical data assimilation (MOMO- Chem) framework" sounds like an inflation of terms. But more importantly, it is mis-leading because the term framework implies to me that the four EnK DA system runs were connected in any way. After reading the paper carefully, I came to the conclusion that this seems not the case. (I hope it did not miss anything). I believe the averaging procedure could be carried for any set of KF DA analysis even without the need to use the same EnKF software. I strongly recommend changing the title and modify the abstract accordingly .*

While we agree that the averaging procedure could in principle be applied to any reanalysis, we disagree that these analyses are disconnected. Modern assimilation systems consist of: (1) model, (2) data, and (3) an estimation algorithm that integrates the two. With the MOMO-Chem framework, components 2 and 3 are the same leaving only component 1 to vary. Consequently, by applying the same data assimilation scheme and assimilating the same sets of observations into the four systems, analysis uncertainty of

individual analyses can be evaluated consistently among models, which have enabled us to provide integrated analysis fields and to quantify forecast model performance on the data assimilation analysis. For typical reanalysis intercomparisons, it is very difficult to disentangle the relative contributions of these components to system performance. To the best of our knowledge, MOMO-Chem provides a unique framework.

There is another way to integrate multi-models in EnKF data assimilation, that combines multi-model forecasts during data assimilation by including them in background error covariances. In this approach, ensembles of models are used to construct flow-dependent analysis system. In the revised manuscript, the following paragraph has been added to explain the possible approaches and the uniqueness of the proposed approach m more clearly:

"Data assimilation that relies on a single model may lead to biased estimation and underestimate model uncertainty by under-sampling of the relevant model space. The limitations with a single model could be overcome by integrating multi-model information in data assimilation in various ways. First, ensembles of models can be used to construct a flow-dependent analysis system. For instance, Xue and Zhang (2014) extended data assimilation to the multi-model Bayesian model averaging analysis framework, in which the posterior model weight for each model is determined through Bayes' theorem reflecting the prior probability of each model and the analysis consistency with the observations. This approach requires a framework to execute and update multiple-model states continuously, which is difficult with multiple state-of-the-art CTMs that have been optimized using different platforms. Another way to integrate multiple-model information is to apply a common data assimilation framework with multiple models. By assimilating the same sets of observations, this framework can be used to demonstrate the importance of forecast model performance on data assimilation analysis, while uncertainty information of individual analyses can be evaluated consistently by using a same data assimilation framework. Uncertainty-weighed multi-model integrated analysis fields would provide unique information that are less dependent on individual model performance and are fundamentally different from averages of individual data assimilation analyses. Quantifying model performance with a multi-model integration is difficult when using different data assimilation frameworks."

With this explanation and the new evaluation results of the multi-model integrated fields (please see our reply below), we think that the current title is appropriate. Meanwhile, the relevant sentences in the abstract and the conclusion have been rewritten to more properly explain our approach as follows:

Abstract:

"We introduce a Multi-mOdel Multi-cOnstituent Chemical data assimilation (MOMO-Chem) framework that directly accounts for model error in transport and chemistry and integrate a portfolio of data assimilation analyses obtained using multiple forward chemical transport models in a state-of-the-art ensemble Kalman filter data assimilation system. The data assimilation simultaneously optimizes both concentrations and emissions of multiple species through ingestion of a suite of measurements (ozone, $NO_2$, CO, $HNO_3$) from multiple satellite sensors."

Conclusion:
"We developed the MOMO-Chem framework to integrate a portfolio of data assimilation analyses obtained using forward CTMs (GEOS-Chem, AGCM-CHASER, MIROC-Chem, MIROC-Chem-H) in a state-of-the-art ensemble Kalman filter data assimilation system."

Furthermore, Fig.1 has been added to explain the general idea of our multi-model data assimilation integration approach using in this study.

 *Combining four analysis using their standard deviation estimate as weights, is of course an advantage of the analyses compared to model runs for which this uncertainty information is not available. On the other hand, the analysis error standard deviation is a derived quantity (from P and R) and not estimated using independent observations. Therefore, it would be beneficial to show that the weighted mean approach leads to better results than an unweighted mean ("multi model"), or perhaps even other approaches to represent ensembles such as the median or the choice of best performing system.*

Thanks for this valuable comment. The comparisons of the uncertainty-weighted and non-weighted multi-model mean have been added in Table 2 and Table 6 in the revised manuscript. The following sentences have been added to explain the results in Section 4.1.2 and Section 4.4. As discussed in the manuscript, the multi-model integration was not applied to CO and NOx concentrations because the analysis spreads of near surface NOx and CO concentrations tend to be similar among the models.

Section 4.1.2:
"The uncertainty-weighed multi-model integrated fields (Eq. 10) show a closer agreement with the ozonesonde observations than the (non-weighted) multi-model means for the lower troposphere, except at SH high latitudes, as summarized in Table 2. The annual and regional mean bias is smaller by 15-40 % in the uncertainty-weighed fields from the SH mid to NH high latitudes, reflecting the larger analysis biases and larger analysis uncertainties in AGCM-CHASER and MIROC-Chem for most cases. The closer agreements suggest improved estimates of ozone in the multi-model integrated fields. In the extratropical

middle and upper troposphere, GEOS-Chem revealed smallest analysis uncertainty and largest analysis errors against the ozonesonde observations, likely associated with the less complex stratospheric chemistry (i.e., smaller spread growth). This model dominated the uncertainty-weighted integrated fields and led to a degradation of the integrated fields. These results suggest a requirement to optimize the analysis uncertainty in some of the systems, considering the fundamental differences in the model framework such as model complexity and resolution, as discussed above. Increasing the number of models would also help to provide more robust statistics."

Section 4.4:

"The integrated analysis shows slightly higher OH concentrations than the multi-model means for most regions, mainly reflecting largest OH spreads and smallest OH concentrations in MIROC-Chem-H among the models. The analysis spread of OH is determined by analysis spreads in various species such as ozone (c.f., Section 3.2) during model forecasts. Because of the different chemical mechanisms and model responses to given perturbations (c.f., Section 5), OH spreads differed by factor of up to 2.5 among the models for the regional means. The integrated uncertainty is smaller than the multi-model spreads by 20-50 % for most regions."

*The validation section (4) only discusses the errors against independent observations of the 4 individual analyses and their unweighted mean ("multi-model") but not the weighted mean ("integrated). This seems not consistent with the intension of the paper and all the discussion before, which focuses on the merits of the weighted-mean ("integrated") analysis. A comparison of unweighted ("multi-model") vs. weighted ("in- tegrated") would be an important aspect of the paper, which is missing. Tables 3 to 6 should also include the respective figures for the weighted ("integrated") means. Further, moving the validation section 4 (Validation results) before section 3 (Data assimilation statistics) seems advisable because having established trust in the data set by the validation could make the discussion on emission response more convincing.*

For the uncertainty-weighted results, please see our reply above.

Because the multi-model integration used in Section 4 is based on the analysis uncertainty shown in Section 3, we think it is straightforward to keep the order. On the other hand, as suggested by the reviewer, it would be better to show the emission response results (Section 3.4) after the validation results (Section 4), because the model response results can benefit from the validation results. Thus, we have moved the emission response section (Section 3.4) to Section 5 (as a new section). Furthermore, the

following sentences have been added to explain the evaluation results of the uncertainty-weighted integrated fields in Section 5 corresponding to the reviewer's comment above:

"Meanwhile, the uncertainty-weighted multi-model integrated ozone fields showed closer agreements with independent observations than the multi-model averages in the lower troposphere (c.f., Section 4.1.2). This suggests that the MOMO-Chem framework provides improved estimates of the atmospheric states for many cases."

*The authors use the standard deviation to represent the variation between the four analyses. Given that 4 is a relatively small number and Gaussianity can not be assumed, using a different measure such as the range should be considered.*

The limitation of this study using the small number of models is discussed as follows in the revised manuscript:
"Given the small number of models (j=1–4) used in this study, the multi-model integration would suffer from sampling biases. With increasing the number of models in future study, this approach would provide more robust statistics."

Any other quantities such as ranges are not added to avoid additional confusions.

*The paper is quite long. Presenting the four reanalysis for concentrations and emission response would be one very valid paper. Comparing the different approaches to combine the reanalyses could be a second paper.*

We think that describing the proposed methodology and presenting the overall performance in a single paper benefits readers. We expect that those readers only interested in specific results, e.g., NOx emissions, will focus on the relevant sections. For the emission response analysis, we plan to conduct a more detailed analysis in a separate paper. Nevertheless, demonstrating the general idea and initial results of the emission response combining in the presented paper is essential to demonstrate the capability of the developed framework.

*Specific comments*

*Abstract:*

*p1 l14. Please consider not to introduce a complex acronym for an averaging procedure of independent data sets.*

Removed the acronyms from the abstract.

*p1 l 21. The sentence starting " These improvements .." is difficult to understand. Improvement would imply a reduction in uncertainty of the emissions but this seems no the case. Please make a quantitative statement on the global or regional total of the prior and posterior emissions.*

The sentence has been rewritten as follow:
"The uncertainty ranges in the a posteriori emissions due to model errors were quantified in 4–31% for NOx and 13–35% for CO regional emissions."

Because of the word limit, we cannot add other information in the abstract. A more quantitative statement is given in the main text.

*p1 l25. The last sentence is not a consequence of the statement before, but quite the opposite. It only shows how difficult the estimation of emissions is because of the complexities of atmospheric chemistry.*

The sentence has been rewritten. Please see our reply below.

*p1l 26. Please clarify "emerging constraints" in the abstract. Also why specifically the "integrated" reanalyses are better suited for this than the single re-analysis or the "multi-model" mean reanalysis.*

The sentence has been replaced as follows to avoid the confusion:
"A systematic investigation of model ozone response and analysis increment in MOMO-Chem could benefit evaluation of future prediction of chemistry-climate system as a hierarchical emergent constraint."

Because there is no space in the abstract and the words "emergent constraints" is becoming well-known, we have added the following sentence in the main text (Section 3.4.1):
"as a hierarchical emergent constraint that uses relationships between future and current climate states to constrain projections of climate response with observations (Bowman et al., 2018). They could also be useful for making effective ozone control strategies."

For the multi-model integrated fields, please see our replies above.

*p2 l20 These are climate runs over several years. In the data assimilation context, model errors are the errors in the forecast over the short assimilation window started from previous analysis. Is not clear that long-term errors are a predictor for the short- range forecast errors of the model. Inconsistencies between the analysis and the model equilibrium state and inadequate background error statistics (inflation factors etc.) may play a large role too, but do not effect the multi-year runs.*

Data assimilation increments can be used to suggest systematic model biases such as in emissions, chemical reaction rate, and deposition rate. These model biases could be regarded as inherent and persistent model errors that could affect long-term model errors. They could partly be measured during a short data assimilation window because of the fast model chemistry and transport processes. By applying data assimilation corrections to these factors, we expect that both short-term and long-term model simulations and data assimilation analyses can be improved to some extent. Note that the relevant discussion is already given in Section 3.4.1 as follows:
"While these simulations describe relatively slow, equilibrium responses, data assimilation incremental updates provide statistics on "fast" responses within the short data assimilation windows. By simultaneously updating ozone and NOx emissions, multi-constituent data assimilation can yield insight into this fundamental quantity."

*p3 l3 "a common data assimilation framework ", see my general comments. I think it is not a common data assimilation framework because the information from the error statistics (P) of each system is not used to come up with a better common analysis. Running different chemistry scheme as part of the ensemble would have been a common framework. The four systems are completely independent. You compare and average different analysis and emissions/concentration responses.*

Please see our replies above.

*p3 l8 Please mention that the assimilated observations are not the same for all 4 reanalyses.*

The sentence has been rewritten as
"Using the same data assimilation settings and assimilating almost the same multi-constituent observations from multiple satellite sensors..."

*p4 l3 Please provide more details on the quantitative value of the inflation factors and any differences between the four applications.*

The sentence has been replaced by

"A covariance inflation factor (Δ, 6% in this study for all the models, following the setting in Miyazaki et al (2015))…"

*P4 l19 "a common data assimilation framework to four CTM frameworks" better "the same data assimilating system using 4 different CTM".*

Rewritten as:

"We applied the same data assimilation system to four CTM frameworks"

*p4 l20 Please include a general statement that the CTMs differ in emissions, chemistry scheme, resolution etc.*

"The major differences among the models are the meteorological input data, the complexity of the chemical mechanisms (simplest in AGCM-CHASER for the troposphere), emission inventories (oldest in GEOS-Chem), vertical coordinate (sigma in AGCM-CHASER only), and spatial resolution (highest in MIROC-CHem-H)."

*p4l 23 - p5 l24 The model descriptions vary to much in detail. In particular section 2.2.4 (MIROC-Chem) seems shorter than the others and does not have detailed information about the emissions. Please avoid too much duplication of information included in the text and provided in table 1. I recommend including the facts in the table and discuss specifics in the text.*

Since the emission data in MIROC-Chem are same as in the AGCM-CHASER, the model description is shorter. Some of the text have been removed to avoid duplicates with the table.

*p7 l3 S and A should be in a bold font as in the formula below*

Corrected.

*p7 l1o Please clarify this sentence. Biases in the observations, in particular if the instrument biases in NO2 and Ozone are not consistent, may severely degrade the performance. I would even speculate the a*

*potential inconsistency between the NO2 and Ozone retrievals are a major reason for problems with the performance of chemical DA systems.*

To explain the importance of bias correction, the sentences have been rewritten as follows:
"Biases in the assimilated satellite retrievals can degrade data assimilation performance. The ozone analysis bias is not solely determined by bias in the assimilated ozone measurements in the multi-constituent data assimilation approach. Miyazaki et al. (2015) demonstrated that the assimilation of measurements other than TES measurements led to corrections in the lower and middle tropospheric ozone. Application of a bias correction procedure for multiple measurements could improve the data assimilation analysis quality. However, we did not apply any bias correction because of the difficulty in estimating the bias structure that could vary temporally and spatially. Meanwhile, since the data is the same for all comparisons with different models, the differences with respect to independent observations are relatively independent of those biases. "

*p8 l11 Please make a comment on the differences between the two MOPITT versions.*

The following sentence has been added:
"The version 7 products have been improved from the version 6 products with respect to overall retrieval biases, bias variability and bias drift uncertainty (Deeter et al., 2017)."

*p9 l2 "We construct integrated data assimilation analysis using multiple models combined with multiple-species measurements." I am not sure if the word "integrated" implies what you do. You calculate a mean of 4 analyses, which were produced with the same DA system using each a different modelling approach. This is not integration or data assimilation as such but simply a weighted average. Please consider replacing the word ("integrated") throughout the text. Please make clear a distinction between the weighted and unweighted mean.*

As discussed in our replies above, we attempted to "integrate" the multi-model data assimilation analyses while considering the analysis uncertainty of individual data assimilation analyses. This provides integrated information on the tropospheric chemistry system. Using the common data assimilation framework and assimilating the same sets of the measurements, this method has enabled us to integrate the multi-model data assimilation analysis fields in a consistent way, which provided fundamentally different information from multi-model means, as demonstrated in the revised manuscript as per the reviewer's commnets.

*p9 l4 Please provide more details on the standard deviation field sigma_j. It would be helpful to include a formula in section 2.1 Also say if sigma_j is a field of the same time-space dimension as the analysis.*

The following sentence has been added:
"The analysis uncertainties are estimated from the root mean square of the analysis ensemble perturbation matrix (see. Eq. (8)) that is obtained by transforming the background ensemble using the local analysis error covariance (see Eq. (5))."

*p9 l5 Please clarify what the spatial resolution of mean reanalysis is, given that the for re-analysis are carried out at different resolutions (table 5)*

The following sentence has been added:
"The multi-model integrated analysis fields were produced at the highest horizontal resolution of the models (1.1°x1.1°) after linearly interpolation."

*p9 l16 Please discuss here (or elsewhere) quantitatively the standard deviations to give an indication which model dominates the weighted-mean analysis.*

The quantitative description of the standard deviations (i.e., analysis uncertainty) is more clearly discussed in the revised manuscript.

*p9 l13 Please clarify why this is not "meaningful"*

The sentence has been rewritten as follow:
"Because of the predefined minimum values of the standard deviations applied to surface emissions of CO and $NO_2$ to prevent covariance underestimation during data assimilation (c.f., Section 2.6), the analysis spreads of near surface NOx and CO concentrations tend to be similar among the models due to the artificial adjustments and are not fully meaningful."

*p9 l26 The longer assimilation window for GEOS-Chem could have implications for the assimilation of NO2 and CO because their life time can be short and the diurnal cycle of the NOx emissions is very pronounced.*

This is already discussed in the manuscript as follow:

"The data assimilation cycle was set to be two hours for the AGCM-CHASER, MIROC-Chem, and MIROC-Chem-H systems, and six hours for the GEOS-Chem system because of the limitation associated with meteorological data input in GEOS-Chem. The emission and concentration fields were analyzed and updated at each analysis step in all the systems. We have confirmed that the results of data assimilation can differ when the data assimilation cycle is changed from two hours to six hours using the AGCM-CHASER system. This occurs, in particular, for the analysis of short-lived species with strong diurnal variability and NOx emission estimates. The performance of the GEOS-Chem data assimilation can thus be expected to differ with the use of a two-hour data assimilation cycle and meteorological data inputs with higher temporal frequency for short-lived species. "

*p11 l20 Please provide evidence for the fact that TES plays the largest role.*

The following sentences have been added:
"Using observing system experiments (OSEs), our previous studies (Miyazaki et al., 2012b, 2015, 2019) revealed that the TES ozone data assimilation dominate the corrections in the tropospheric ozone analysis, whereas the use of measurements other than TES measurements (mainly $NO_2$ measurements) led to corrections in the lower and middle tropospheric ozone during the forecast."

*p12 l24 instead of "integration" use "mean"*

Please see our replies above.

*p12 l31 The smaller standard deviation seems "trivial" given the formula of its calculation. Only the comparison with independent observation can demonstrate that the mean is a more reliable representation of the field.*

The results of the multi-model integrated analysis are compared with independent observations in the revised manuscript. Please also see our replies above.

*p13 l16 Please provide more detail how the regression coefficients are calculated. Do they represent the mean over regions, month and all hours of the day? Please explain how the response is calculated for the weighted mean ("integrated") analysis.*

The sentence has been rewritten as follows:

"By applying linear regressions to the multi-model integrated fields (c.f., Section 3.3), we evaluated model responses of surface ozone and $NO_2$ concentrations to NOx emissions. We first produced the daily multi-model integrated fields at 1.1° x 1.1° resolution and then applied linear regressions."

*p13 l29 Please clarify that sentence. The study comes up with a different response functions for each model but gives no guidance which is the correct one. Given the non-linearities a mean over conflicting responses might not be the most suited approach.*

Because there is no truth information (i.e., observational estimates) on the responses, we cannot provide such guidance. The sentence addresses that the MOMO-Chem provides a framework to quantify the response of forward models and its multi-model differences, without making any sensitivity calculations. So, its principal value is a diagnostic quantity. Future research can further assess whether one or more of these responses is more valid—and their implications for assessing the impact of emission changes.

*p14 l17 I would think "conflicting" is a better description then "unique" for the difference in the response*

The MOMO-Chem multi-model integrated fields provides different estimates of the responses from the individual models. Thus, "unique" is a reasonable statement.

*p15 l13 Please clarify" inter-model correlations"*

Rewritten as "The correlations of delta ENOx and delta ozone among the models estimated at each point at each day..."

*p15 l29 Please clarify if the only difference between "mean" and "integrated" (formula 10) is the weighting with the analysis error standard deviations. If this is larger than the unweighted mean, this means a model with a larger "ozone response" had a smaller analysis error standard deviation. In this sense the standard deviation of the analysis error is also used as an indicator for the reliability of the "ozone response". Is this true an how can this be justified?*

We estimated the response using the multi-model integrated fields (integrated) and compared them with the averages of the individual response estimates (mean). These can be different because of the non-Gaussian distributions of the individual model fields. The procedure is more clearly described in the revised manuscript as follows:

"We first produced the daily multi-model integrated fields at 1.1° x 1.1° resolution and then applied them to linear regressions."

"Finally, the model responses differ significantly between the MOMO-Chem integrated fields (solid blue and red lines) and the mean of the individual model estimates obtained by averaging the model responses from individual model fields"

*p11-p18 Please discuss also the errors of the weighted mean ("integrated") analysis and update the tables*

Tables 3 and 6 have been updated to include uncertainty-weighted integrated values and its uncertainty. Please also see our replies above.

*p18 l10 Please clarify this sentence. What are the estimated mean errors (the biases?) and how do they relate to the analysis uncertainty, which is expressed as (bias corrected) standard deviation*

Thank you for the comment. I agree with the reviewer. The sentence has been rewritten as
"The estimated RMSEs (2.5-9.0 ppbv at the SH high latitudes, 3.0-4.3 ppbv at the SH mid latitudes, 2.5-5.3 ppbv in the tropics, 0.7-3.8 ppbv at the NH mid latitudes, and 2.6-6.3 ppbv at the NH high latitudes for 850-500 hPa) are significantly smaller than the analysis uncertainty (Fig. 1b) in AGCM-CHASER and MIROC-Chem (10–16 ppb) and is comparable to that in GEOS-Chem and MIROC-Chem-H."

*p18 l13. Are you suggesting the AGCM-CHASER and MIROC-Chem should be given more weight in the weighted ("integrated") analysis.*

Yes. Note that changes to analysis uncertainty should affect analysis mean.

*p22 l22 The reduced multi-model spread for OH is an important finding and should presented more prominently. It shows that the assimilation of O3, NO2 and CO can improve the OH field because different models come to the similar results even without assimilation of OH observations.*

To emphasize the results, the following sentence has been added:

"The obtained OH fields, which are less dependent on individual model performance due to reduced model errors in relevant species, demonstrate the potential of the multi-constituent (ozone, CO, and $NO_2$) data assimilation for various atmospheric chemistry studies including emission inversion and methane budget analyses."

*p23 l11 Please give here also the respective numbers for the prior emissions.*

Added.

*p23 l20 The comparison seems not quite correct. The prior emissions are biased against each other, which will also strongly influence the posteriori spread.*

To discuss the a priori and a posteriori emission spread relationship more carefully, the sentence has been rewritten as follows:
"The multi-model spread of the a posteriori regional NOx emissions is smaller than the assumed a priori emission uncertainty (i.e., by 40 %) for all the polluted areas (Table 8), while the a priori emission spreads could influence the obtained a posteriori emission spreads. From sensitivity calculations, we confirmed that the daily emission updates greatly reduce the dependence of the a priori emissions for many regions (now shown)."

*p25 l5 Could these rather large and diverse correction of the CO emission indicate that the important impact of the VOC emissions on CO concentrations is not considered with the setup of EnKF DA system?*

That is possible. To discuss its possibility more clearly, the relevant sentence has been rewritten as follows:
"Also, differences in the chemical production of CO from the oxidation of NMHCs and the chemical lifetime of CO, which were not optimized by the data assimilation, could lead to large multi-model discrepancies in CO simulations and emission estimates, as similarly discussed by Gaubert et al. (2016)."

*p28 l2 Please mention that these results were obtained by a regression approach.*

Added.

*p27l18 Please clarify uncertainty ranges in this context. Does it included that the biases or is it just the standard deviation. You should mention that the prior emissions had smaller differences between each other than a posteriori emissions.*

The sentence has been rewritten as
"...which are quantified in 4–31% for NOx and 13–35% for CO regional emissions from a multi-model spread of the a posteriori emissions."

The prior emissions do not show smaller differences for some cases, which are clearly shown in Table 8 in the revised manuscript. Please also see our reply above.

*p28 l5 "The obtained results also suggest that the multi-model integrated fields could provide fundamentally different chemical relationships than those in the individual models" I find this statement not convincing. A divergent response of the different model shows predominately that a robust conclusion can not be drawn from this multi-model approach. If only, it means we need more research to find more realistic chemical relationships.*

To emphasize the importance of further research, the following sentence has been added:
"Meanwhile, more research is needed to comprehend detailed chemical mechanisms."
Meanwhile, the given sentence is reasonable because the multi-model integrated fields clearly provide fundamentally different information from individual models.

*Table 8 Please include multi-model mean and spread for prior emissions Caption: Please indicate what the number in brackets are*

Added.

*Figure 3: Indicate what the dashed lines represent Figure 8: Correct spelling of "deveation"*

Indicated.

*Figure 15: The differences between the different a posteriori and prior estimates between the models could be shown better.*

The multi-model spread of the a posteriori emissions are already shown in Figure 17. Detailed spatial patterns for the a priori emissions do not provide any useful information.